# Charge-density wave mediated quasi-one-dimensional Kondo lattice in stripe-phase monolayer 1T-NbSe$_2$

Zhen-Yu Liu[1,5], Heng Jin[2,3,5], Yao Zhang[1,5], Kai Fan[1,5], Ting-Fei Guo[1], Hao-Jun Qin[1], Lan-Fang Zhu[1], Lian-Zhi Yang[1], Wen-Hao Zhang[1], Bing Huang[2,3] ✉ & Ying-Shuang Fu[1,4] ✉

The heavy fermion physics is dictated by subtle competing exchange interactions, posing a challenge to their understanding. One-dimensional (1D) Kondo lattice model has attracted special attention in theory, because of its exact solvability and expected unusual quantum criticality. However, such experimental material systems are extremely rare. Here, we demonstrate the realization of quasi-1D Kondo lattice behavior in a monolayer van der Waals crystal NbSe$_2$, that is driven into a stripe phase via Se-deficient line defects. Spectroscopic imaging scanning tunneling microscopy measurements and first-principles calculations indicate that the stripe-phase NbSe$_2$ undergoes a novel charge-density wave transition, creating a matrix of local magnetic moments. The Kondo lattice behavior is manifested as a Fano resonance at the Fermi energy that prevails the entire film with a high Kondo temperature. Importantly, coherent Kondo screening occurs only in the direction of the stripes. Upon approaching defects, the Fano resonance exhibits prominent spatial 1D oscillations along the stripe direction, reminiscent of Kondo holes in a quasi-1D Kondo lattice. Our findings provide a platform for exploring anisotropic Kondo lattice behavior in the monolayer limit.

Kondo lattice arises from the hybridization between a matrix of periodically arranged localized magnetic moments and an itinerant electron bath, resulting in low-energy excitation with heavy electron mass, i.e. heavy fermion systems[1–3]. The interplay between on-site Kondo screening and inter-site exchange coupling leads to various intriguing many-body quantum phenomena[4–6], ranging from non-Fermi liquid[7,8], quantum criticality behavior[9–11], and unconventional superconductivity[12–14] to topological correlated states in combination with spin-orbital interaction[3,15,16]. Such subtle mutual interactions impose challenges for understanding those fruitful quantum phases. One approach to simplify the problem is to start with one-dimensional (1D) Kondo lattice models, which can be strictly solved to obtain the ground-state phase diagram[17,18]. Moreover,

unconventional quantum criticality, involving Kondo destruction at the quantum critical point, is expected to occur at reduced dimension[19–21], rendering 1D Kondo lattice systems important in their own light. However, the existing heavy fermion materials are mostly 3D $f$-ion compounds[1,12] with rare systems of quasi-1D character[21–23].

Recent advances in van der Waals (vdW) materials have extended Kondo lattice systems to $d$-electron systems[24–27], which have revealed new phenomena not seen in conventional $f$-electron systems, e.g., stacking-order-dependent hybridization[26], ferromagnetism-promoted heavy fermion states[24], and intercalation tunable of Kondo screening[27]. Nevertheless, as one of the most classical model systems, (quasi-) 1D Kondo lattice materials are still lacking. A prerequisite for that purpose is to generate matrixes of spin chains. The introduction of line defects

[1]School of Physics and Wuhan National High Magnetic Field Center, Huazhong University of Science and Technology, Wuhan, China. [2]Department of Physics, Beijing Normal University, Beijing, China. [3]Beijing Computational Science Research Center, Beijing, China. [4]Wuhan Institute of Quantum Technology, Wuhan, China. [5]These authors contributed equally: Zhen-Yu Liu, Heng Jin, Yao Zhang, Kai Fan. ✉e-mail: Bing.Huang@csrc.ac.cn; yfu@hust.edu.cn

to monolayer VSe$_2$ has proven to be a viable way of creating ferromagnetism in the otherwise nonmagnetic vdW crystal[28]. In this regard, inducing line-defects mediated magnetism in metallic systems potentially offers a promising route to realize the long-sought 1D Kondo lattice behavior.

In this work, we achieved the growth of monolayer NbSe$_2$ in the stripe phase, and unveiled its Kondo lattice behavior using spectroscopic imaging scanning tunneling microscopy (SI-STM). The stripe phase in monolayer NbSe$_2$ is caused by Se-deficient line defects generated during molecular beam epitaxy (MBE) growth, which leads to a charge density wave (CDW) phase transition forming triple period superstructure along the stripe direction and opens a CDW gap centered slightly below the Fermi energy. The local magnetic moment held by each CDW unit cell is revealed by first-principles calculations and found to couple with the itinerant CDW band via SI-STM, leading to quasi-1D Kondo lattice behavior. Furthermore, we observed a real-space oscillation of the Kondo resonance width adjacent to defects with four times the CDW period, suggesting the behavior of Kondo holes in a quasi-1D Kondo lattice[29,30].

## Results

The stripe phase monolayer NbSe$_2$ films were grown by MBE on graphene-covered 4H-SiC(0001) substrate at elevated substrate temperature through fine tuning of the Se:Nb flux ratio (Fig. 1a–c). With a high Se:Nb flux ratio, the NbSe$_2$ films are grown into a 1 T structure with a $\sqrt{13} \times \sqrt{13}$ pattern of CDW (Fig. 1a), known as the star of David (SD) motifs. The SD phase is a charge-transfer insulator[31]. Under a low Se:Nb flux ratio, the films are grown into the stripe phase (Fig. 1c). A medium Se:Nb flux ratio results in coexisting stripe phase and SD phase (Fig. 1b). Interestingly, the two phases can be interconverted in a reversible manner by annealing the preformed films with different conditions. Namely, the 1 T phase can be transformed into the stripe phase with vacuum annealing, and further be restored upon annealing the stripe phase under Se flux (Fig. 1a–c, Supplementary Fig. 1).

STM topography of the stripe phase indicates the stripe direction is along the island edge, at an angle of 13.9° relative to the SD direction of the 1 T phase (Fig. 1b). There are two types of stripes that have different widths, i.e., wide stripes (~1.26 nm) and narrow stripes (~0.95 nm). As is seen from Fig. 1d, narrow strips dominate the films in stripe phase, which are imbedded with sparse wide stripes. Both stripes display a period of ~1.04 nm along the stripe direction, which is three times the constant of the 1T-NbSe$_2$ lattice (0.344 nm)[32] (Fig. 1d). For neighboring stripes, the modulation phase along the stripe shifts by a half period, thus forming a triangular pattern. As will be shown later, this triangular pattern is a newly discovered CDW phase.

Both the stripe phase and the 1 T phase can appear on the same NbSe$_2$ island and have the same layer height (Fig. 1b, Supplementary Fig. 2), suggesting their intimate connection. In view that the stripe phase has Se deficiency during growth, it is natural to expect that this stripe phase is associated with Se-deficient line defects caused by the low Se flux. Such conjecture is validated from atomic resolution STM imaging of the stripe phase, which displays periodically arranged trenches spanning the stripes with single columns of Se atoms located on the trenches (Fig. 1e). The narrow and wide stripes contain two and three columns of Se atoms, respectively, where the CDW pattern with a triple period along the stripes is evidently seen. While surface Se atoms appear intact all over the surface, the Se atoms in the trenches are about 1 Å deeper than those between the trenches, suggesting the existence of line defects at the trenches (Fig. 1f, Supplementary Fig. 3).

Next, we focus on the narrow stripes and perform tunneling spectroscopy measurement at 2 K to characterize their electronic structure. (The spectra of wide stripes are shown in Supplementary Fig. 4). The d$I$/d$V$ spectrum features a spectral dip centered at -0.1 eV, below which a broadband appears with two peaks at approximately -0.3 eV and -0.55 eV (Fig. 2a). Above the spectral dip, there is another broad band surpassing the Fermi level with two peaks at 0.05 eV and 0.15 eV. Above 0.3 eV, the spectral density monotonically decreases and reaches nearly zero at 0.8 eV, above which the spectral intensity

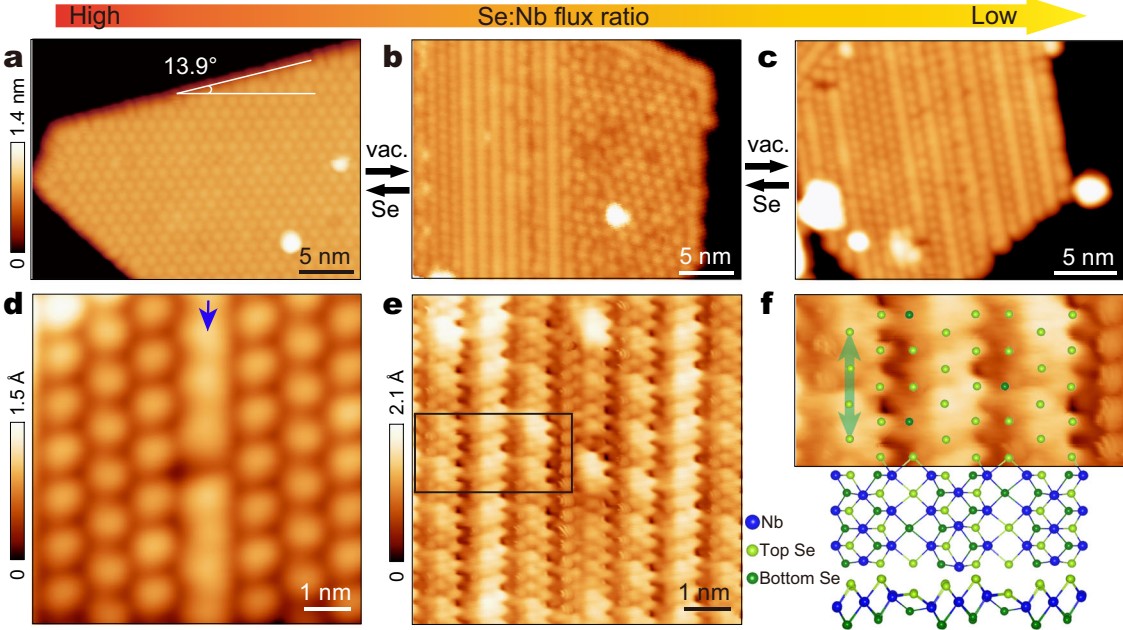

**Fig. 1 | Growth and morphology of stripe-phase monolayer NbSe$_2$. a–c** STM images ($V_b = -1.0$ V and $I_t = 10$ pA) showing morphology of the grown 1T-NbSe$_2$ films in SD phase **a**, mixed phase **b**, and stripe phase **c**. The SD phase and the stripe phase can be interconverted with annealing under vacuum and Se flux conditions, respectively. **d** Enlarged STM image ($V_b = -0.7$ V and $I_t = 60$ pA) of the stripe-phase. The blue arrow marks a wide stripe. **e**, Atomic resolution STM image ($V_b = 20$ mV and $I_t = 3$ nA) of the same area as **d**. **f** Zoom-in STM image of the stripe-phase NbSe$_2$ extracted from the black rectangle in **e** and superimposed with its DFT-calculated structure, whose top (up) and side (down) views are shown. For clarity, only protruded Se-layer atoms are overlaid on the image. The green line segment marks the CDW periodicity along the stripe.

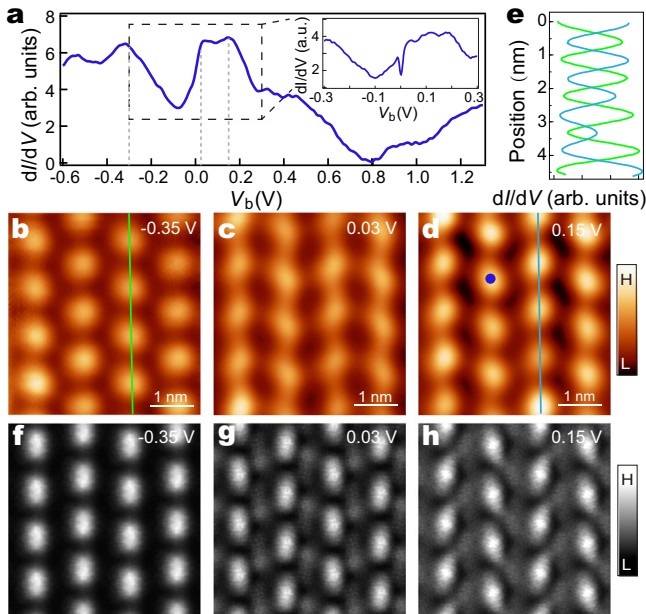

**Fig. 2 | d$I$/d$V$ spectrum and conductance maps of the narrow stripe. a** d$I$/d$V$ spectrum ($V_b = -0.6$ V, $I_t = 200$ pA, $V_{mod} = 20$ mV) taken at the blue dot in **d**. The inset shows a zoom-in d$I$/d$V$ spectrum ($V_b = -0.3$ V, $I_t = 100$ pA, $V_{mod} = 5$ mV) around the Fermi level. **b–d** Constant-height d$I$/d$V$ maps of the same area at characteristic energies (for 0.03 V, $V_{mod} = 5$ mV, the others $V_{mod} = 20$ mV). **e** Line profiles of the d$I$/d$V$ intensity along the green **b** and blue **d** line, respectively. **f–h**, DFT-calculated d$I$/d$V$ maps at the corresponding energies.

grows again with increasing energy. The d$I$/d$V$ mapping of the narrow strips shows the triangular pattern with a triple period along the stripes for all the energies, whose modulation phase, however, varies prominently with energy (Fig. 2b–d, see Supplementary Figs. 5 and 6 for more details). Specifically, the d$I$/d$V$ mapping exhibits enhanced intensity at the stripes but with an evident anti-phase spatial modulation for energies below and above the spectral dip center, for instance -0.35 eV and 0.15 eV, respectively (Fig. 2e). Moreover, for energies above 0.8 eV, another spectral dip center, the conductance intensity becomes enhanced along the trenches (Supplementary Fig. 6). The triple period conductance modulation and its anti-phase relation at both sides of the spectral dip around the Fermi level hallmarks a CDW state, where anti-phase charge modulations occur at both edges of the CDW gap[33,34].

Having identified the CDW state, we scrutinize the fine spectral features around the Fermi level. As shown in the inset of Fig. 2a, the d$I$/d$V$ spectrum exhibits a pronounced sharp dip, which is ascribed as an anti-resonance, superimposed on the higher CDW band. This sharp anti-resonance appears within a very narrow energy window of about 20 meV and right at the Fermi level, suggesting its origin as a many-body resonance state. The d$I$/d$V$ map at 0.03 eV (Fig. 2c), corresponding to the energies close to the anti-resonance state, shows that the spatial distribution of the anti-resonance state consists with that of the upper CDW band at 0.15 eV, implying the correlation between the two collective states. While the unit shape of the CDW distribution is more rounded, that of the anti-resonance distribution is elliptical.

We focused on the narrow spectral range around the Fermi level and investigated the nature of the anti-resonance state. A superconducting gap with nodes could exhibit a "V" shaped spectrum. However, neither vortex states nor change in the spectra dip is detected under magnetic field (Supplementary Fig. 7), excluding such a scenario. The disorder-induced pseudo gap can also be excluded[35], because the stripe-phase NbSe$_2$ has a well-defined stoichiometry

without prominent disorder, as is evidenced from the ordered spectroscopic mappings in Fig. 2b–d. Another possibility of the sharp dip is a spin excitation in view of its narrow dip width. Indeed, a careful inspection reveals the spectral dip is asymmetric with an adjacent hump at lower energy (Fig. 3a). This asymmetric spectral shape is reminiscent of the Fano line shape commonly found in Kondo resonance systems[36,37]. As such, we fitted the spectrum with a Fano line shape superimposed with a linear background. The fitting nicely matches the experimental spectrum, which gives a Kondo peak width $\Gamma$ of 7.9 meV and a Fano factor of -0.4 (Fig. 3a, red curve). The Fano factor is determined by the two interfering tunneling paths from the STM tip into the Kondo state and the itinerant electrons, where the Kondo dip-like spectrum reflects the latter path dominates. The Kondo resonance is further justified from its temperature evolution, where the sharp dip broadens rapidly with increasing temperature (Fig. 3a, blue curves). Such spectral dips at elevated temperatures are shallower than the numerically calculated spectra merely from the influence of Fermi-Dirac distribution based on that measured at 2 K (Fig. 3a, gray dashed curves), demonstrating the spectral dip becomes intrinsically suppressed with elevated temperature. We used a thermally convolved Fano function with a linear background to fit the spectra at different temperatures [Fig. 3a, red curves] and extracted $\Gamma$ (Fig. 3b, red dots). The relation of $\Gamma(T)$ can be excellently fitted using the well-established expression for Kondo systems, i.e. $\Gamma(T) = \sqrt{2(k_B T_k)^2 + (\pi k_B T)^2}$, rigorously proving its Kondo origin. The fitting yields a Kondo temperature $T_k \sim 64$ K [Fig. 3b, blue curve], which is significantly higher than that in vdW 1T/1H-TaS$_2$[26]. In our system, the localized moments and the itinerant electrons are both inside the same monolayer, makes their relative tunneling ratio sensitive to local perturbations. Indeed, the Fano spectrum in proximity to certain regions with inhomogeneity exhibits a sharp Kondo peak with similar $\Gamma(T)$ relation as the Kondo dip, and a Kondo gap with two enhanced peaks at gap edges (Supplementary Fig. 8). Those spectra could be reproduced well by the cotunneling model, that has been well established for the spectra of Kondo lattice system[38,39], with different tunneling ratios between the Kondo state and the itinerant bands (Supplementary Fig. 8d, see details of the model in Supplementary Note 1). Interestingly, the DFT calculations also reveal that the tunneling amplitude to the itinerant bands could be sensitive to strain (Supplementary Note 2).

To evaluate the spatial dependence of the Kondo resonance, we selected a region that is free from adjacent defects, and acquired line spectra surpassing multiple CDW units at 2 K. As shown in Fig. 3c, e, the spectra along and traversing the stripes all display Kondo resonances, evidencing the Kondo resonance is distributed all over the entire film. The global nature of the Kondo state indicates the formation of Kondo lattice, where each CDW unit of the stripe phase hosts a localized magnetic moment. This resembles the case of SD motifs in the CDW pattern of 1T-TaS$_2$, where a localized spin resides at each SD center[26]. The fitted Kondo resonance widths and amplitudes along the stripe are reasonably uniform (Fig. 3d), demonstrating formation of the coherent Kondo screening[40]. However, traversing the stripes, although the Kondo peak width fluctuates moderately, the Kondo amplitude has obvious oscillations with the same periodicity as the stripes (Fig. 3f). The maximum Kondo amplitudes slightly shift from the centers of the CDW units (Fig. 3f), conforming to the locations of the calculated local moments. This implies the Kondo coherence happens dominantly along, instead of traversing, the stripes, which signifies the quasi-1D character of the Kondo lattice system.

To further elucidate the electronic properties of the CDW phase and the origin of the localized magnetic moments, we carried out DFT calculations (See method). As seen in Fig. 1f, the atomic resolution STM image of the stripe-phase resembles that of VSe$_2$[28,41]. As such, we first adopted the same structural model in our case as VSe$_2$. However, the STM images cannot be reproduced (Supplementary Fig. 9). Therefore,

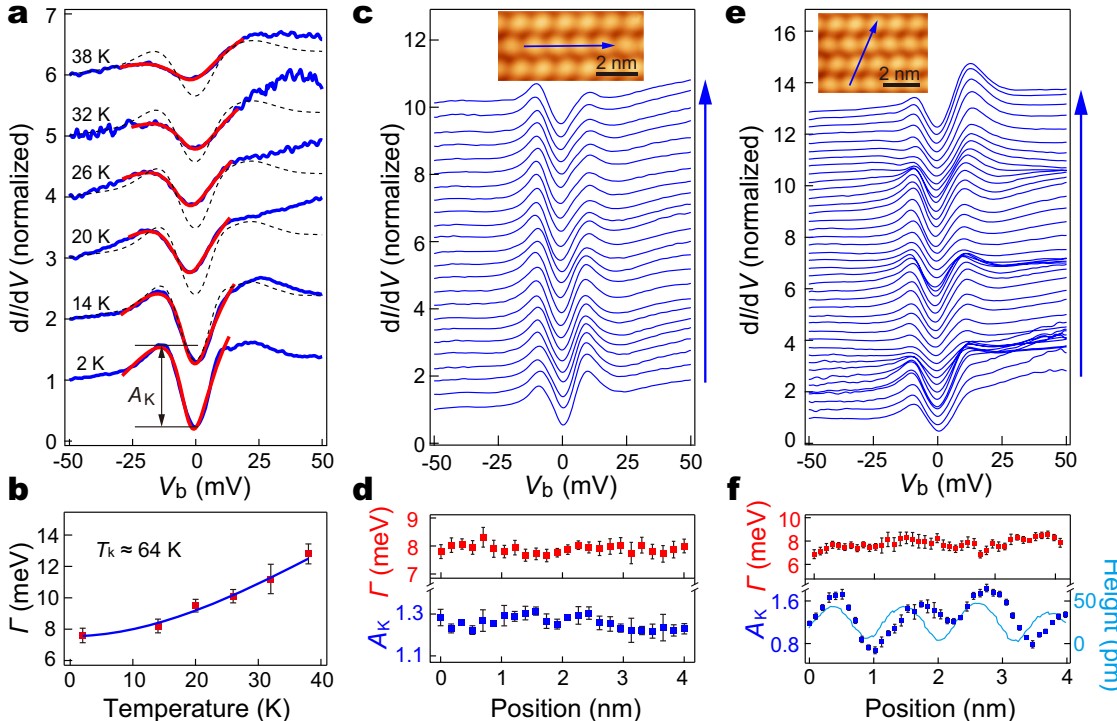

**Fig. 3 | Temperature and spatial dependences of the Kondo state. a** d$I$/d$V$ spectra (blue curves) measured at different temperatures on the same location of stripe-phase NbSe$_2$ ($V_b = -50$ mV, $I_t = 200$ pA, $V_{mod} = 0.5$ mV). Red curves are fittings to the resonance with a Fano function at each temperature. Black dotted curves are numerically calculated spectra based solely on the thermal broadening of the spectrum at 2 K. The spectra are vertically offset for clarity. **b** The extracted Kondo peak width ($\Gamma$) (red dots) against temperature. The blue curve is a fitting to the data according to the Kondo model, yielding a Kondo temperature of

$T_k \approx 64$ K. **c, e** Waterfall plot of the spatially distributed d$I$/d$V$ spectra ($V_b = -100$ mV, $I_t = 200$ pA, $V_{mod} = 0.5$ mV) along the blue arrows in their respective inset STM images ($V_b = 0.3$ V and $I_t = 20$ pA). The stripes are along the horizontal direction. **d, f** Kondo resonance widths (red dots) and amplitudes (blue dots) of the d$I$/d$V$ spectra extracted from **c** to **e** at different locations, respectively. The height profile in **f** is taken along the blue line in the inset image of **e**. The error bars in **b, d,** and **f** are from the Fano fitting.

different atomic configurations of undistorted and CDW phases are constructed. The undistorted structure, namely the primitive cell, is characterized by every three Nb lines divided by one Se line, as shown in Fig. 4a. Phonon spectrum in Fig. 4b suggests that major structural instabilities are located at $1/2\mathbf{b_1} + 1/3\mathbf{b_2}$ and $1/3\mathbf{b_2}$ respectively ($\mathbf{b_1}$ and $\mathbf{b_2}$ are the reciprocal lattice vectors). Especially, the instability at $1/2\mathbf{b_1} + 1/3\mathbf{b_2}$ implies the structural reconstruction in a $2 \times 3$ supercell, which agrees with the period of the observed stripe-phase. The CDW phase transition is accompanied by the movement of Nb atoms along the stripe direction (see red arrows in Fig. 4a) and the sinking of certain Se atoms in the Se lines (see Fig. 4a, c). In the view of electronic properties, the primitive cell is spin-unpolarized, but the formation of the CDW phase generates spin polarization. The spin density distribution (Fig. 4c) clearly shows major magnetic moments located at the more isolated Nb atoms rather than in the center of Nb clusters (brown hexagon) in the CDW phase, forming a matrix of ordered local magnetic moments. The matrix has the same period as the CDW pattern, providing the basis of the Kondo lattice. The conducting electrons around the Fermi level are mainly contributed by Nb $4d$ orbitals close to the Se defect line (Supplementary Fig. 10), forming a quasi-1D conducting channel. Besides, as shown in Fig. 4d, e, the band renormalization opens a full (partial) gap in the spin-up (spin-down) channel below the Fermi level in the formation of the CDW phase. The strong reduction of calculated total DOS, i.e., CDW gap, is mainly centered at -0.25 eV, slightly deviating from the experimental dip at -0.1 eV, which may be partially caused by the charge transfer between the film and the substrate or strain induced by sample inhomogeneity (Supplementary Fig. 11). Importantly, considering the substrate-induced Fermi level shift (Fig. 4d, e), the anti-phase charge modulations associated with

the CDW phase transition are observed at both sides of the CDW gap (see Fig. 2f-h), agreeing with experimental d$I$/d$V$ mappings (see Fig. 2b–d).

Two evidences from DFT calculations can strongly illustrate the quasi-1D nature of this stripe CDW phase. The first one is that the Fermi surface of the CDW phase is found to be parallel-like lines rather than closed curves in both spin channels (Fig. 4f), indicating that the CDW system is quasi-1D like rather than 2D like. The second evidence is that the calculated exchange interactions between nearest magnetic moments on Nb atoms (see Supplementary Fig. 12) show that magnetic interactions are rather anisotropic, as usually found in lower-symmetry two-dimensional magnetic systems[42,43]. As shown in Fig. 4c, the interaction along the stripe directions ($J_1$) is about two times of that between different stripes ($J_2$ and $J_3$), i.e., strong (weak) magnetic interaction along (perpendicular to) the stripe direction. By projected bands with magnetic orbitals (Supplementary Fig. 13), we find that the local moments in the stripe-CDW phase are comparable with the SD-CDW phase, and the bandwidth ratio of the dispersive band and flat band in the stripe phase is comparable with the typical $d$ electron Kondo lattice system LiV$_2$O$_4$[44,45]. Although these $d$ bands that contribute to the major localized magnetic moments are far below the conducting bands holding itinerant electrons (Supplementary Fig. 14) in terms of their energy level positions, these moments can still be effectively screened by the itinerant conduction electrons via spin-flip scattering, which involves a virtual hopping process[46]. Interestingly, the calculated magnetic interaction in this stripe-CDW phase is about two orders of magnitude stronger than that in the SD-CDW phase, possibly due to the stronger interaction between local moments and background conducting electrons.

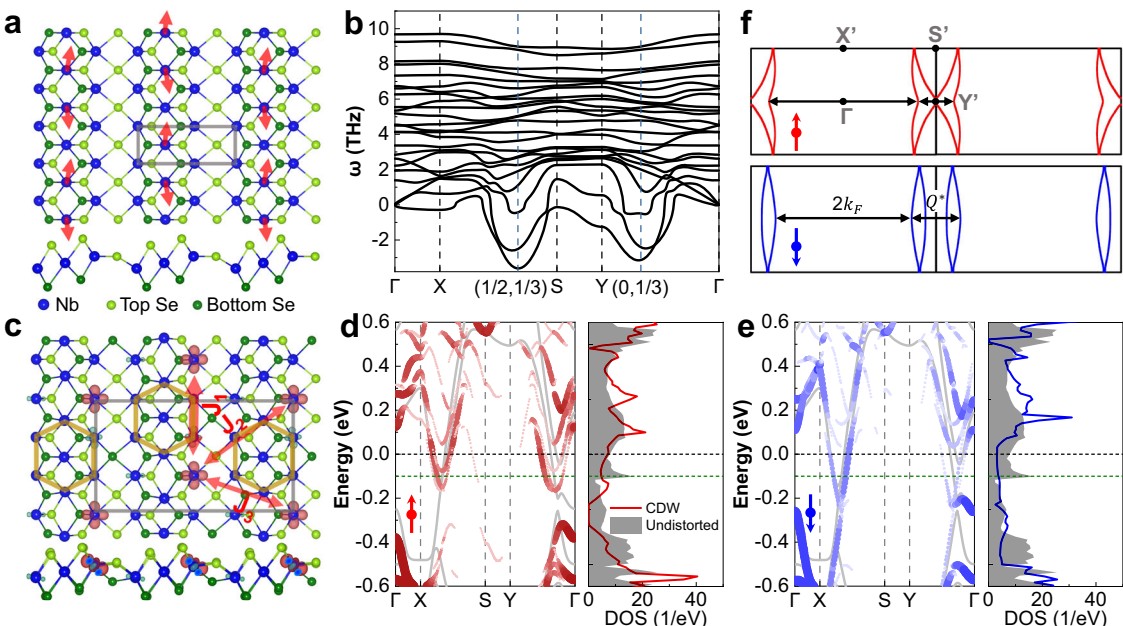

**Fig. 4 | DFT-calculated structural and electronic properties of undistorted and CDW phases. a** Atomic configuration of the undistorted phase. The primitive cell is illustrated with a gray rectangle. Red arrows indicate major movements of Nb atoms during the CDW formation. **b** Phonon spectrum of the undistorted phase. X, Y, and S are high-symmetry points in the first Brillion zone of the undistorted phase. **c** Atomic configuration and spin density distribution of the CDW phase. Spin density is shown with 0.006 e/Å³ isosurface. The nearest magnetic exchange interaction along the stripes ($J_1$) and that between the neighboring stripes ($J_2$ and $J_3$) are illustrated with red bidirectional arrows. The thickness of the arrows indicates

the strength of the magnetic exchange interaction. The clustering of Nb atoms during the CDW phase transition is illustrated with brown hexagon. **d**, **e** Unfolded band structures and density of states (DOS) of the CDW phase in spin-up (red) and spin-down (blue) channel, respectively. The band structures and DOS of the undistorted phase are also plotted (gray) for comparison. While black-dashed lines indicate the Fermi level of the free-standing CDW phase, the green-dashed lines indicate the downshift of the Fermi level induced by the graphene substrate. **f** Spin-resolved Fermi surface of the CDW phase. X′, Y′, and S′ are high-symmetry points in the first Brillion zone of the CDW phase.

To further examine the 1D Kondo lattice behavior, we investigated the spatial dependence of Kondo parameters near defects, that are generated during the film growth. The defect appears as depressions at the CDW units (Fig. 5a–c, inset). At the defect site, the Kondo resonance changes substantially (Supplementary Fig. 15), suggesting the defect may act as a Kondo hole, which arises from a non-magnetic impurity imbedded inside the Kondo lattice[29]. The $dI/dV$ spectra taken across several CDW units along the stripe at 2 K indicate the Kondo resonance changes prominently (Fig. 5d–f). The parameter $\Gamma$ of the Kondo resonances, which are extracted from the Fano line fitting of the spectra, surprisingly exhibits pronounced spatial oscillations (Fig. 5a–c). The oscillations decay from defects in a pattern described by an exponentially decayed sinusoidal function: $\Gamma(r) = Ae^{-\sigma r}\sin(\omega r + \varphi) + b$. The amplitude of this pattern decays with a typical length of ~8.3 ± 0.6 nm, and the period of oscillation is ~3.9 ± 0.6 nm (about four times the CDW period). These properties are consistent for all three data sets, whose statistics give the above uncertainty. Furthermore, the oscillation is symmetric on both sides of the defects, as shown in Supplementary Fig. 16. The spatial oscillation of $\Gamma$ is consistent with the Kondo hole, which expects oscillation in the hybridization strength between the spin lattice and the itinerant electrons. The periodicity of the oscillation has a wave vector of $2k_F^c$ in classical heavy fermion systems, where $k_F^c$ is the Fermi wave vector of the light band of the conduction electrons[29,30]. As shown in Fig. 4f, our DFT calculation gives $2k_F^c \approx 0.75$ reciprocal lattice units (r.l.u.), which corresponds to a wave vector $Q^*$ of about 0.25 r.l.u. in the extended zone scheme[25], conforming to the experimentally observed oscillation period in real space. We have also attempted to investigate the oscillation of $\Gamma$ at directions traversing the stripes. However, the films are suffered from their limited number of neighboring narrow stripes that are interrupted

with coexisting wide stripes, preventing such measurements. Nevertheless, the anisotropic feature of the stripe phase and the strong oscillation of $\Gamma$ along the stripe near the Kondo hole all characterize the Kondo lattice of 1D in nature. The Kondo resonance width in our case is too large to induce an observable Zeeman splitting under the available magnetic field strength. Nevertheless, overall, based on (i) the spatial oscillation of the Kondo linewidth, (ii) the changes of Kondo lineshape reproduced with the cotunneling model, (iii) the temperature-dependent evolution of the Kondo lineshape, the Kondo lattice behavior can be well identified, making alternative charge- or spin-ordering scenarios unlikely (Supplementary Note 3).

In summary, we have achieved the growth of monolayer NbSe₂ in the stripe phase using MBE. Utilizing SI-STM, we unveiled a new type of CDW order and, more importantly, a zero-bias anomaly that prevails the film surface, signifying the formation of a Kondo lattice. Those observations are substantiated by DFT calculations, which suggest the CDW order coexists with an ordered lattice of local moments, as well as the 1D character of the electronic states. The Kondo lattice indicates prominent oscillation upon approaching defects along the stripes, pointing to the behavior of Kondo holes imbedded inside a quasi-1D Kondo lattice. In view that 1D Kondo lattice model can be strictly solvable and its expected unique quantum critical behavior, the monolayer stripe-phase NbSe₂ represents an unprecedented 1D Kondo lattice system, that provides a benchmark for examining the novel heavy fermion physics. This finding not only enriches the Kondo lattice systems in $d$-electrons, but also extends heavy fermions to the monolayer limit. This system envisions extensive in-depth studies, since monolayer vdW crystals are readily tunable with external means, such as electrostatic gating, optical fields, surface doping, and the construction of vdW heterostructures, etc.

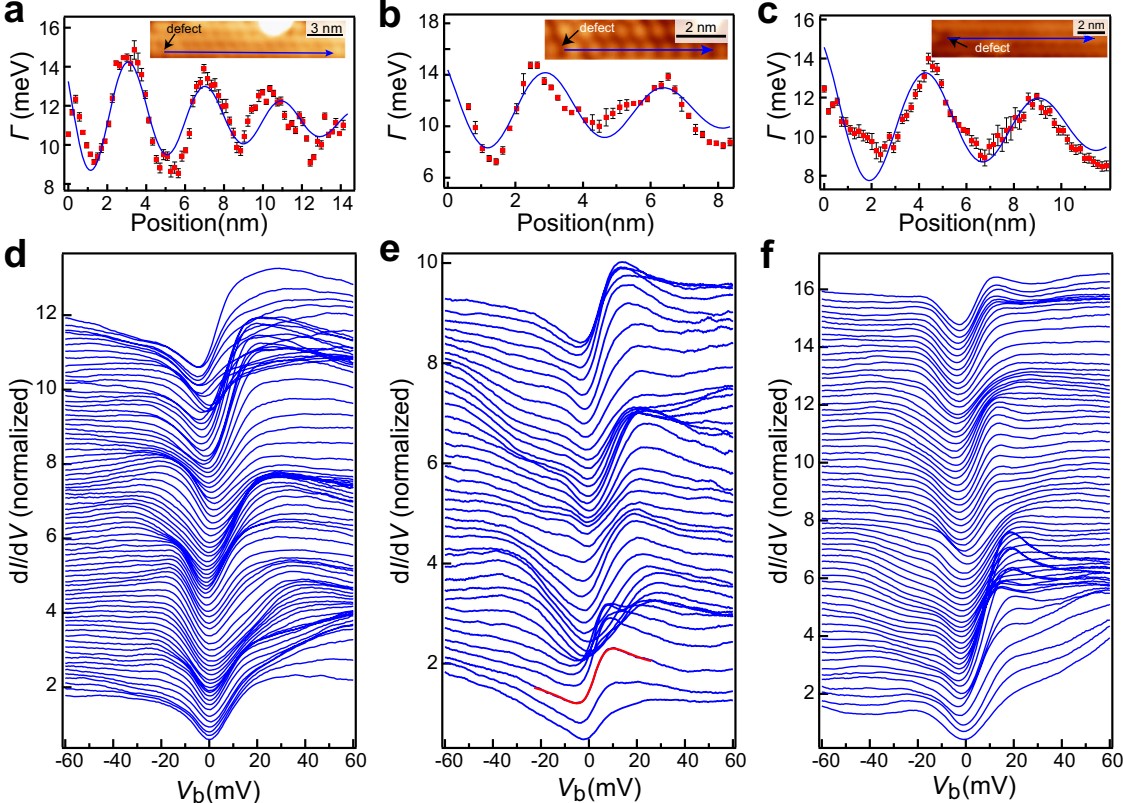

**Fig. 5 | Spatial oscillation of the Kondo resonance adjacent to defects.**
**a–c** Spatial oscillation of the Kondo peak width $\Gamma$ (red dots) near defects along the strip direction, that are extracted by fitting the d$I$/d$V$ spectra in **d–f** respectively. The blue curves are fittings to the data with an exponentially decayed sinusoidal function. Inset STM images show the stripe phase NbSe$_2$ with marked defects (black arrow) and the spectral locations (blue arrow). **d–f** The d$I$/d$V$ spectra taken along the blue arrows in the inset images of **a–c**. The red curve in **e** shows a typical Fano fit to the spectrum.

## Methods

### Sample growth

The stripe phase NbSe$_2$ was grown on epitaxial graphene on 4H-SiC(0001) substrate by MBE in an ultrahigh-vacuum (UHV) chamber (base pressure of $5 \times 10^{-10}$ mbar). High-purity Nb (rod, purity: 99.85%) and Se (shot, purity: 99.999%) were evaporated from a home-made electron beam evaporator and a home-made Knudsen cell evaporator, respectively. The film growth rate is ~0.005 monolayer/min and the substrate was maintained at 600 °C during the growth. 1T-NbSe$_2$ and stripe phase NbSe$_2$ were grown by increasing or decreasing the flux of Se, respectively. The coexisting phase was obtained by a medium Se flux, and then annealed at 600 °C for 30 min to obtain high-quality monolayer stripe phase NbSe$_2$ for experimental measurements.

### STM measurements

The STM measurements were carried out with a custom-designed Unisoku STM (1500). All the measurements were taken at 2 K, unless specifically specified. An electrochemically etched W tip was first calibrated on the Ag(111) film grown on Si(111) substrate before the measurements. The d$I$/d$V$ spectra were acquired by standard lock-in technique at a frequency of 973 Hz. The d$I$/d$V$ maps was performed in the constant-height mode.

### First-principles calculations

Our first-principles calculations were performed with Vienna ab initio simulation package (VASP)[47] where the projector augmented wave method was adopted[48]. The exchange-correlation term was treated with the Perdew-Burke-Ernzerhof (PBE) form of the generalized gradient approximation[49]. The $4p^6 4d^4 5s^1$ and the $4s^2 4p^4$ electrons were treated as valence electrons for Nb and Se, respectively. A vacuum layer of about 15 Å was added to avoid interactions between periodical layers. Gaussian smearing of 0.05 eV was used in our calculations. Energy cutoff of 400 eV and k mesh of $4 \times 10 \times 1$ and $2 \times 3 \times 1$ in BZ were adopted for structural relaxation of the undistorted phase and CDW phase, respectively, with force on each atom less than 0.002 eV/Å and the convergence criterion of total energy being $10^{-6}$ eV. Increased $k$ mesh of $8 \times 20 \times 1$ and $4 \times 6 \times 1$ over BZ and convergence criterion of total energy $10^{-7}$ eV were adopted in static calculations of undistorted and CDW phases respectively. We adopted the LDA + $U$ scheme[50] with $U_{\mathrm{eff}} = 3$ eV to Nb $4d$ electrons[51], which gave the best description of experimental d$I$/d$V$ mapping among $U$ varying from 2 eV to 4 eV. The simulation of d$I$/d$V$ maps was based on the summation of DOS at corresponding energy $E_b$ within range $E_b \pm 0.01$ eV and a distance of about 4.5 Å from the surface. The effect of spin orbital coupling (SOC) was negligible in our system and not considered. The calculations of the phonon spectrum were based on a $6 \times 2 \times 1$ supercell of undistorted phase with density functional perturbation theory (DFPT) as implemented in PHONOPY software[52]. The spin susceptibility was evaluated with the help of the WANNIER90 software package[53]. The Fermi surface was visualized with FermiSurfer software[54] and the unfolded band structure was calculated with BANDUP code[55,56].

## Data availability

The data that support the findings of this study are available from the corresponding author upon request.

## Code availability

The code that supports the findings of this study is available from the corresponding author upon request.

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

## Acknowledgements
We thank Yifeng Yang, Testuo Hanaguri, Yongkang Luo for the discussions. This work was funded by the National Key Research and Development Program of China (Grant Nos. 2022YFA1402400), the National Natural Science Foundation of China (Grant Nos. 92265201, U20A6002, 12088101, 12047508), and NSAF under Grant No. U2230402.

## Author contributions
Y.S.F. conceived the project. Z.Y.L., Y.Z., and K.F. performed the experiments with the help of T.F.G., H.J.Q., L.F.Z., L.Z.Y., W.H.Z. H.J. and B.H. carried out the DFT calculations. Z.Y.L., Y.Z., K.F., H.J., B.H., and Y.S.F. analyzed the data. Y.S.F., B.H., Z.Y.L., Y.Z., and H.J. wrote the manuscript with comments from all authors. Y.S.F. and B.H. supervised the project.

## Competing interests
The authors declare no competing interests.
