## [Peer Review File · Nature Communications]

Reviewers' comments:

Reviewer #1 (Remarks to the Author):

In this manuscript, Zhen-Yu Liu et al. report a combined study of STM and DFT calculations about the stripe-phase monolayer NbSe₂. The authors first prepare the stripe-phase monolayer NbSe₂ by MBE growth. Then they observe a sharp dip-like feature near the Fermi level in the dI/dV spectra taken on the stripe-phase monolayer NbSe₂. The authors fit the dip-like feature with Fano function, and they conclude that it is a Kondo resonance. Furthermore, the authors also find that the Kondo resonance feature shows spatial oscillation near the atomic defect. However, I think the authors did not provide the smoking-gun evidence about the Kondo resonance feature. Except the dip-like feature near the Fermi level, no other novel phenomena, such as quantum criticality behavior or unconventional superconductivity is observed in the stripe-phase NbSe₂. I think this work does not meet the criteria of Nature Communications. I do not recommend the publication of this manuscript in Nature Communications. Below are my detailed comments.

1. As reported in Ref. 41, when monolayer 1T-NbSe₂ is grown on HOPG, there is a Mott insulating gap. Did the authors also observe the insulating gap in the SD-phase 1T-NbSe₂ shown in Fig. 1a? Why the stripe-phase monolayer NbSe₂ grown on graphene/SiC shows the “Kondo” resonance rather than the insulating gap? In this system, where do the conducting electrons for screening the local moments come from?
2. Although the authors fit the sharp dip-like feature in the dI/dV spectra taken on stripe-phase NbSe₂ with the Fano function, I am not convinced that it is a Kondo resonance. First, there are often some dip-like features near the Fermi level for TMDs materials, such as in monolayer 1T-TiSe₂ [Science Bulletin 63, 426 (2018)] and 1H-TaSe₂ [Nano Letters 18, 689 (2018)]. I think these dip features can also be more or less fitted by the Fano functions, but they are not Kondo resonance. Second, this sharp dip-like feature has almost no response to external magnetic field (Fig. S6). There are many possibilities for the dip-like feature near the Fermi level, such as phonon excitation and pseudo gap. Since the Kondo resonance is the key point for this manuscript, the authors have to show the smoking-gun evidence for the Kondo resonance nature of the dip-like feature.
3. For the monolayer 1T-TaS₂ and monolayer 1T-NbSe₂ system (Refs. 26 and 41), both show the strong Kondo resonance peak. Why there is dip-like feature in the strip-phase 1T-NbSe₂?
4. From the dI/dV linecut profiles shown in Fig. 2e, the min and max points in the blue and green lines do not exactly match with each other, and there is some phase shift. What is the reason for this phase shift?
5. I don't think their DFT calculations match with the experimental data in this manuscript. First, the charge transfer effect between monolayer TMDs material and graphene/SiC is usually negligible [Nature Physics 16, 218 (2020)]. Why there are strong charge transfer effect between graphene and

stripe-phase NbSe₂ (the Fermi level needs to be shifted by 150 mV)? If the authors include the graphene substrate in the DFT calculations, is there also strong charge transfer effect between graphene and stripe-phase NbSe₂ in the calculations? Second, in order to have the localized spin to form Kondo resonance, there should be a flat band located near the Fermi level. But why there is no clear flat band feature near the Fermi level in the DFT calculations (Figs. 4d and 4e)?

6. Near the atomic defects, the authors observe the spatial oscillation of the Kondo resonance width with about four times the CDW period. Instead of only showing the oscillation behavior for one atomic defect in Fig. 4, I think the authors should investigate more this kind of atomic defects (at least four or five defects), and the Kondo resonance for all the defects should show spatial oscillations with the same or similar period.

7. The authors claim that the 1D Kondo lattice attracts special attention because of its exact solvability, can the 1D Kondo lattice for the stripe-phase NbSe₂ system be exactly solved? Why there are no other novel properties observed in this Kondo lattice system, such as quantum criticality behavior or unconventional superconductivity?

Reviewer #2 (Remarks to the Author):

In this paper reported by Liu et al, a new-type of CDW order coexisting with a quasi-1D Kondo lattice behavior was unveiled in monolayer NbSe₂ with Se-deficient line defects utilizing scanning tunneling microscopy. The Kondo behavior manifests as a Fano resonance around the Fermi level, which further exhibits spatial 1D oscillations along the stripes. First-principles calculations suggested that the localized magnetic moment held by each CDW unit cell leads to the formation of the quasi-1D Kondo lattice. These findings strongly indicate an unprecedented 1D Kondo lattice model in a d-electron system, which might also extend the understanding of the heavy fermion physics in the monolayer limit. I'd like to suggest publication after the minor technique issues are properly addressed.

1. From Figure 1, two types of stripes with different widths can be observed. Do the wide stripes exhibit similar 1D Kondo lattice behavior as the narrow stripes?

2. Could the authors comment on what roles played by the Se-deficient line defects in the formation of the new CDW order and thus 1D Kondo lattice?

3. Did the authors consider more atomic models for the 1D Se-deficient line? For example, there is at least another model proposed by Liu et al (Nano Lett., 2019, 19, 4897) for the 1D Se-deficient line in monolayer VSe₂.

Reviewer #3 (Remarks to the Author):

The manuscript by Liu et al. presents spectroscopic investigation of the novel metallic stripe phase in NbSe₂ monolayer grown with molecular beam epitaxy under reduced Se flux. The authors focus on its most salient feature – narrow dip in the vicinity of the Fermi level – its depth and width exhibit intriguing quasi-1D spatial dependence. With the help of DFT calculations, this feature is ascribed to the Kondo state. The authors, thus, suggest that Se-deficient stripes in NbSe₂ could be the first experimental realization of the archetypal 1D Kondo model, where a variety of exotic excitations has been predicted.

The manuscript attempts to address challenging yet highly rewarding (if successful) problem of emergent magnetic orders in non-magnetic TMDC monolayers and its heterostructures. It clearly demonstrates novel and intriguing results, but I am not convinced that the evidences in favor of a Kondo state are compelling. I believe the manuscript is not suitable for publication in Nature Communications, but the material science aspect – stripe phase growth and characterization – is clearly suitable for a regular publication elsewhere.

1. My major concern is the lack of clear identification of localized magnetic moments. The authors suggest that ‘more isolated Nb atoms’ could act as such moments based on DFT calculations. However, neither of predictions of these calculations are subject to further scrutiny or experimental testing. Moreover, I can clearly see that the bands in spin-resolved theoretical spectrum are rather dispersive, with small effective mass not immediately suggestive of localization.
2. The assignment of the spectral dip to Kondo state is premature, in my opinion. For example, spin-density wave order on a part of (spin-polarized!) Fermi surface pockets could account for a temperature-dependent ‘pseudo’-gap. Interestingly, the ratio of the gap to apparent T_c is within the expected range.
3. Moreover, non-stoichiometric composition of the monolayer and potentially random positions of missing Se could suggest that the introduced disorder could facilitate the disorder-induced localization forming a pseudogap (see e.g. Gao et al., Nano Lett. 2020, 20, 9, 6299–6305), completely unrelated to Kondo physics. Neither of the above possibilities (2,3) were ruled out.

4. Use of contrast inversion as a hallmark of CDW remains controversial (see e.g. Spera et al., PRL 125, 267603 (2020) and discussions therein). Further evidence is highly desirable to support the CDW rather than a structural origin of the reconstruction within the chain.

Reply to Review

Reviewer #1 (Remarks to the Author):

In this manuscript, Zhen-Yu Liu et al. report a combined study of STM and DFT calculations about the stripe-phase monolayer NbSe₂. The authors first prepare the stripe-phase monolayer NbSe₂ by MBE growth. Then they observe a sharp dip-like feature near the Fermi level in the dI/dV spectra taken on the stripe-phase monolayer NbSe₂. The authors fit the dip-like feature with Fano function, and they conclude that it is a Kondo resonance. Furthermore, the authors also find that the Kondo resonance feature shows spatial oscillation near the atomic defect. However, I think the authors did not provide the smoking-gun evidence about the Kondo resonance feature. Except the dip-like feature near the Fermi level, no other novel phenomena, such as quantum criticality behavior or unconventional superconductivity is observed in the stripe-phase NbSe₂. I think this work does not meet the criteria of Nature Communications. I do not recommend the publication of this manuscript in Nature Communications. Below are my detailed comments.

Thank the referee for taking the time reviewing our manuscript. The dip-like feature is ascribed to a Kondo anti-resonance from its temperature evolution that is expected for the thermal broadening effect of the Kondo resonance. To further justify its Kondo origin, we have provided additional data showing change in Fano line shape at selected region with sample inhomogeneity. Despite that no quantum criticality or unconventional superconductivity is discovered, we believe that identifying a quasi-1D Kondo lattice system in a monolayer van der Waals crystal is certainly a significant finding, which already meets the high standard of Nature Communications, as are also recognized by the other two reviewers. The referee's comments are constructive, which certainly help to improve our manuscript. Taking those comments, we have made substantial changes to our manuscript with additional experimental and theoretical data added, which rigorously support our conclusions. The referee's support of publication in Nature Communications would be greatly appreciated. Our point-by-point responses to the referee's issues are below.

1. As reported in Ref. 41, when monolayer 1T-NbSe₂ is grown on HOPG, there is a Mott insulating gap. Did the authors also observe the insulating gap in the SD-phase 1T-NbSe₂ shown in Fig. 1a? Why the stripe-phase monolayer NbSe₂ grown on graphene/SiC shows the "Kondo" resonance rather than the insulating gap? In this system, where do the conducting electrons for screening the local moments come from?

Thanks for this issue. As is correctly pointed out by the referee, there is a Mott gap in monolayer 1T-NbSe₂ on HOPG. In our study, we used graphene-covered SiC(0001) substrate, whose electronic properties are similar to HOPG. We did observe the charge-transfer insulating gap in monolayer SD-phase 1T-NbSe₂, and the relevant results have been reported in our previous work in Ref. [Nano Lett. 2021, 21, 16, 7005–7011].

The stripe-phase NbSe₂ contains regularly spaced Se-deficient line defects, which profoundly change the stoichiometry of the compound to Nb₃Se₅. As a result, its electronic structure changes distinctly different to the SD-phase. The conducting electrons around Fermi level for screening the

local moments is mainly contributed by Nb $4d$ orbitals close to the Se defect line (Fig. S10), as has been depicted in Page 10 of the main text.

For clarity, we have added a sentence “The SD phase is a charge-transfer insulator [31]” to Page 4.

2. Although the authors fit the sharp dip-like feature in the dI/dV spectra taken on stripe-phase NbSe_2 with the Fano function, I am not convinced that it is a Kondo resonance. First, there are often some dip-like features near the Fermi level for TMDs materials, such as in monolayer 1T-TiSe_2 [Science Bulletin 63, 426 (2018)] and 1H-TaSe_2 [Nano Letters 18, 689 (2018)]. I think these dip features can also be more or less fitted by the Fano functions, but they are not Kondo resonance. Second, this sharp dip-like feature has almost no response to external magnetic field (Fig. S6). There are many possibilities for the dip-like feature near the Fermi level, such as phonon excitation and pseudo gap. Since the Kondo resonance is the key point for this manuscript, the authors have to show the smoking-gun evidence for the Kondo resonance nature of the dip-like feature.

Thanks for raising this critical issue. The evidences for the Kondo resonance are from the following aspects: (1) The dip-like feature can be fitted with Fano resonance. The resonance width is only about ~ 8 meV at 2 K, which is typical for magnetic excitations. (2) Our DFT calculations suggest the existence of spin lattice, which is responsible for the Kondo resonance. The validity of our calculations is confirmed from the excellent match between the calculated and experimental STM images in Fig. 2 b-h. (3) The evolution of the resonance width Γ increases rapidly with elevating temperature T , whose relation could be fitted well by the Kondo relation $\Gamma(T) = \sqrt{2(k_B T_k)^2 + (\pi k_B T)^2}$ (see Fig. 3b of the main text). This can be considered as a smoking-gun evidence for the Kondo resonance. It is noted that similar temperature evolution behavior has been used to justify the Kondo resonance showing spectroscopic dips in several other Kondo lattice systems (e.g., see Refs. PNAS 107, 10383 (2010), Nature 474, 362 (2011), Sci. Adv. 4, eaao6791 (2018)). Actually, in single impurity Kondo effect such as Ref. Phys. Rev. Lett. 88, 077205 (2002), the evidences for the Kondo resonance have been identified from the existence of local moment, Fano line fitting, and the temperature evolution relation. (4) Near defects, the dip-like feature indicates spatial variations in its resonance width with a period corresponding to the Fermi wave length of the conduction electrons. This signifies the feature of Kondo hole, providing another evidence for the Kondo resonance. (5) The Kondo resonance should have a Fano line shape, defined by the q -factor, as is determined by the relative tunneling ratio between the two paths into localized Kondo state and itinerant electrons. In our system, the localized moments and the itinerant electrons are both inside the same monolayer. This makes the ratio between the two tunneling paths, namely the q factor of the Kondo resonance, sensitive to local perturbations. As such, we have measured the spectra in proximity to some region with inhomogeneity (please see Fig. R6 and our reply to the second question of Referee 3). We found the spectrum exhibits a sharp resonance peak at the Fermi level, whose temperature dependent behavior conforms nicely to a Kondo peak, thus unambiguously justifying its Kondo origin. All these (1)-(5) evidences undisputedly point to the existence of Kondo lattice in our system.

As is correctly pointed out by the referee, there are dip features near Fermi level for CDW gaps existing in many TMD materials such as monolayer 1T-TiSe_2 [Science Bulletin 63, 426 (2018)] and 1H-TaSe_2 [Nano Letters 18, 689 (2018)]. However, those systems do not contain spin moments, and no temperature evolution of its dip width is identified for a Kondo relation, differing from our case.

It is indeed that our Kondo resonance has almost no response to magnetic field. That is understandable, because the Kondo resonance width is 8 meV, which is nearly 8 times larger than the Zeeman splitting energy of ~ 1 meV induced by a magnetic field of 10 T with a g factor of 2.

While phonon excitations and pseudo gaps could also exhibit the dip like feature, their spectral shapes are not typically of asymmetric Fano line shapes, and not possible to exhibit peak-like feature in any sample regions either.

For clarity, we have made the following changes: (1) A phrase “in view of its narrow dip width” is added to Page 7, and a phrase “rigorously proving its Kondo origin” is added to Page 8; (2) A sentence “The Kondo resonance is not split at 8 T because its Zeeman splitting energy is way smaller than its Kondo peak width.” is added to the caption of Fig. S7; (3) A sentence starting with “In our system, ...” is added to Page 8.

3. For the monolayer 1T-TaS₂ and monolayer 1T-NbSe₂ system (Refs. 26 and 41), both show the strong Kondo resonance peak. Why there is dip-like feature in the strip-phase 1T-NbSe₂?

For STM measurements, the tunneling spectrum of a Kondo lattice should be manifested as a Fano line shape “because of the presence of two interfering tunneling paths from the STM tip, one directly into the itinerant electrons, and the other indirectly through the Kondo resonance” (see the quoted description in Ref. PNAS 107, 10383 (2010).) Thus, the Kondo resonances at different heavy fermion systems show different Fano shapes, because the above mentioned tunneling channels vary. For instance, in heavy fermion compound URu₂Si₂, the Kondo resonance has an asymmetric Fano line shape (Ref. Nature 465, 570 (2010)), which is similar to our case. In another heavy fermion system CeCoIn₅ (Ref. Nature 486, 201 (2012)), the Kondo resonance appears as a dip on Co-terminated surface and a peak on Ce-In-terminated surface, respectively. This is because the two surfaces have different ratio of electron tunneling into the itinerant electrons and Kondo resonances.

In the cases of monolayer 1T-TaS₂ and monolayer 1T-NbSe₂, their spin lattices are in the top 1T layer and are screened by itinerant electrons from the metallic substrate underneath. As a result, STM tunneling through the Kondo resonance dominants, making their Kondo resonances exhibiting as peaks. In our case, the localized spin lattice and the itinerant electrons are in the same monolayer, and our STM tunneling is dominated with delocalized itinerant electrons, and thus our Kondo resonance manifests itself as dip-like features.

For clarity, we have added a sentence “The Fano line shape is determined by two interfering tunneling paths from the STM tip into the Kondo state and the itinerant electrons, where the dip-like Kondo spectrum reflects the latter path dominates.” to Page 7.

4. From the dI/dV linecut profiles shown in Fig. 2e, the min and max points in the blue and green lines do not exactly match with each other, and there is some phase shift. What is the reason for this phase shift?

Thanks for raising this issue. We carefully examined the constant-height dI/dV maps at different energies and found that the map at -0.35 V has a stricter contrast reversal relation than the original -0.3 V, and have thus updated Figs. 2b and f with the images at -0.35 V. The line profiles are also updated in Fig. 2e, showing exactly matched inverse relation.

5. I don't think their DFT calculations match with the experimental data in this manuscript. First, the charge transfer effect between monolayer TMDs material and graphene/SiC is usually negligible [Nature Physics 16, 218 (2020)]. Why there are strong charge transfer effect between graphene and stripe-phase NbSe₂ (the Fermi level needs to be shifted by 150 mV)? If the authors include the graphene substrate in the DFT calculations, is there also strong charge transfer effect between graphene and stripe-phase NbSe₂ in the calculations? Second, in order to have the localized spin to form Kondo resonance, there should be a flat band located near the Fermi level. But why there is no clear flat band feature near the Fermi level in the DFT calculations (Figs. 4d and 4e)?

We thank the referee for this good comment, inspiring us to think more about the origin of Fermi level shift. First, as mentioned by the referee, the charge transfer effect between monolayer TMD and graphene/SiC is usually negligible, also evidenced by our DFT calculations for SD-phase NbSe₂ on monolayer graphene. As shown in Fig. R1(a), only slight electrons are transferred from graphene substrate to the upper Hubbard band of the SD phase. As concerned by the referee, a charge transfer by substrate may not account for the shift of Fermi level of ~ 150 mV. Here, we suggest that this shift could be caused by the joint effect of substrate-induced strain and charge transfer effects. For example, when a ~3% strain is induced to match the stripe phase and the graphene substrate lattice, as shown in Fig.R1(b), a shift as large as ~100 mV is obtained (see the arrows in the figure), suggesting that the influence of substrate on the stripe-phase can be large and shift the Fermi level towards the experimental value.

For clarity, we have added Fig. R1 to supplementary information as Fig. S11. A sentence "which may be partially caused by the charge transfer between the film and the substrate (Fig. S11)" is added into Page 10.

Fig. R1. **a**, Substrate-induced charge transfer effect on SD-phase NbSe₂. **b**, joint strain- and substrate-induced a large energy shift in stripe-phase NbSe₂.

Second, as for the localized spin, we are sorry to cause such impression that no flat band is observed in the Fig. 4d and 4e. The reason could be two-fold. One is due to the use of unfolded band structure in the figures. The other is due to the energy range in the figures is too small to well describe the localized magnetic moments. As shown in Fig. R2(a), where the main orbital $d_{x^2-y^2}$ contributing to the magnetic moment of magnetic Nb in the stripe-phase is projected to the band, the bandwidth of which is ~ 0.35eV, comparable with bandwidth of d_{z^2} orbital of Nb in the center

of SD phase [Fig. R2(b)], which is considered to present local moments. In Fig. R2(c), we highlight the possible conducting electron channel of opposite spin of stripe-phase, with bandwidth of $\sim 0.7\text{eV}$. We note that though for the stripe-phase, the bandwidth ratio of “flat band” and “dispersive band” ($\sim 1/2$ in the NbSe₂ stripe-phase) are much larger than typical heavy fermion systems with f electrons, e.g., SmB₆ [Acta Physica Polonica A, 2014, 126(1): 298-299]. it’s still comparable with LiV₂O₄, the first Kondo lattice system with d electrons, whose bandwidth of the “flat band” is $\sim 1\text{eV}$ while bandwidth the of “dispersive band” is $\sim 2\text{eV}$ [Phys. Rev. B 60, 16359 (1999), Phys. Rev. Lett. 83, 364 (1999)].

For clarity, we have added Fig. R2 to supplementary information as Fig. S13. A sentence “By projected bands with magnetic orbitals, we find that the localization of magnetic moments in stripe-CDW phase is comparable with SD-CDW phase (Fig. S13), and the bandwidth ratio of dispersive band and flat band in stripe phase is comparable with typical d electron Kondo lattice system LiV₂O₄” is added into Page 11.

Fig. R2. Comparison of band structure of NbSe₂ stripe-phase and SD-phase. **a**, spin-up and **c**, spin-down bands of stripe-phase. **b**, spin-up and **d**, spin-down bands of SD-phase. Orbitals mainly contribute to the magnetic moments ($d_{x^2-y^2}$ for stripe-phase and d_{z^2} for SD-phase) are projected and plotted with yellow dots. In **c** the thickened line indicates possible dispersive band coupled to local moments. The indication of bandwidth is shown with shadowed window.

6. Near the atomic defects, the authors observe the spatial oscillation of the Kondo resonance width with about four times the CDW period. Instead of only showing the oscillation behavior for one atomic defect in Fig. 4, I think the authors should investigate more this kind of atomic defects (at least four or five defects), and the Kondo resonance for all the defects should show spatial oscillations with the same or similar period.

Following the reviewer's suggestion, we have investigated more this kind of defects in the stripe-phase monolayer NbSe₂. Fig. R3 shows another three data sets of dI/dV spectra measured along the stripes, which, in conjunction with Fig. 5, constitute four data sets in total. Obviously, all the three data sets in Fig. R3 exhibit spatial oscillations in Kondo resonance widths along the direction of the stripe away from the defects. The oscillation periods are all kept around 4 nm in all the data sets. For clarity, we have added Fig. R3 to supplementary information as Fig. S15.

Fig. R3. a, c, e, Spatial oscillation of the Kondo peak width Γ (red dots) near defects along the strip direction, that are extracted by fitting the dI/dV spectra in b, d, f, respectively. Inset STM images show the stripe phase NbSe₂ with marked defects (black arrow) and the spectral locations (blue arrow). b, d, f, The dI/dV spectra taken along the blue arrows in the inset images of a, c, e. The red curve in d shows a typical Fano fit to the spectrum.

7. The authors claim that the 1D Kondo lattice attracts special attention because of its exact solvability, can the 1D Kondo lattice for the stripe-phase NbSe₂ system be exactly solved? Why there are no other novel properties observed in this Kondo lattice system, such as quantum criticality behavior or unconventional superconductivity?

Thanks for raising this issue. In heavy fermion physics, the 1D Kondo lattice model has strict solvability with numerical methods. It is noted that such exact numerical solvability requires input parameters of single impurity Kondo coupling strength and the inter-spin exchange coupling strength, which need to be precisely determined in the stripe-phase NbSe₂ with more extended studies such as ARPES and transport. To tune the system to quantum critical points, one needs to use external fields such as strain or electrostatic doping, etc, which are beyond the scope of current

study. Unconventional superconductivity does not always occur in heavy fermion systems but appear around quantum criticality if any.

Reviewer #2 (Remarks to the Author):

In this paper reported by Liu et al, a new-type of CDW order coexisting with a quasi-1D Kondo lattice behavior was unveiled in monolayer NbSe₂ with Se-deficient line defects utilizing scanning tunneling microscopy. The Kondo behavior manifests as a Fano resonance around the Fermi level, which further exhibits spatial 1D oscillations along the stripes. First-principles calculations suggested that the localized magnetic moment held by each CDW unit cell leads to the formation of the quasi-1D Kondo lattice. These findings strongly indicate an unprecedented 1D Kondo lattice model in a d-electron system, which might also extend the understanding of the heavy fermion physics in the monolayer limit. I'd like to suggest publication after the minor technique issues are properly addressed.

Thank the referee for the positive and encouraging comments on our study. The issues raised by Referee 2 are very helpful to improve our manuscript. Below are our point-by-point responses to the comments.

1. From Figure 1, two types of stripes with different widths can be observed. Do the wide stripes exhibit similar 1D Kondo lattice behavior as the narrow stripes?

Thanks for this question. Fig. R4 shows spectra of wide stripes. Its large energy range spectrum is distinct to that of the narrow stripes, and no Kondo spectrum is observed around the Fermi level. For clarity, we have added Fig. R4 to supplementary information as Fig. S4.

Fig. R4. **a**, STM image ($V_b = -0.6$ V and $I_t = 10$ pA) of the wide stripe. **b**, **c**, dI/dV spectra with different energy range taken at the location marked with red dot in **a**. No Kondo spectrum is observed from **c**.

2. Could the authors comment on what roles played by the Se-deficient line defects in the formation of the new CDW order and thus 1D Kondo lattice?

Thanks for this comment. The role of Se-deficient line could be described with three aspects. First, without the deficient line, the pristine 1T-NbSe₂ would form SD phase. Taking the line defect into account, the three-fold rotational symmetry is broken, resulting in a rectangular primitive cell, forbidding the formation of SD phase. Second, due to lack of Se, the Fermi surface is strongly changed, thus new electronic instability could be found, possibly related to different Fermi surface nesting apart from the 1T phase. Third, the Se-deficient lines result in larger distance between the Nb-stripes, reducing the interactions between stripes, benefiting the 1D character of the system.

3. Did the authors consider more atomic models for the 1D Se-deficient line? For example, there is at least another model proposed by Liu et al (Nano Lett., 2019, 19, 4897) for the 1D Se-deficient line in monolayer VSe₂.

Thanks for mentioning this article. So far, as far as we know, only two atomic models are proposed for the 1D Se-defect line in VSe₂, while the other one [*Adv. Mater.* **32**, 2000693 (2020)] has been discussed in our original manuscript. We've considered whether this atomic model [*Nano Lett.*, 2019, 19, 4897] for the 1D Se-deficient line in monolayer VSe₂ could also relate to the NbSe₂ stripe phase. As shown in the Fig. R5(a) and (b), after geometric relaxation in the DFT calculations, we obtained a similar crystalline structure as in VSe₂ stripe system. However, the lattice constant obtained within this model is rather large (~1.12 nm), about ~18% larger than the experimental value (~0.95 nm). Based on this atomic model, a 2 × 3 CDW phase with local magnetic moments could be obtained, shown in Fig. R5(c) and (d). However, the lattice constant of this CDW-mediated stripe phase doesn't vary much compared with the pristine phase, unlikely to fit the experimental value. Though this atomic structure does not match experimental data, we've added it in Fig. S9 in the revised supplementary information for comparison.

Fig. R5. **a** (top view) and **b** (side view) of atomic model of pristine NbSe₂ stripe-phase in agreement with the reference (Nano Lett., 2019, 19, 4897). **c** (top view) and **d** (side view) of spin density distribution of a 2 × 3 CDW-mediated stripe-phase. The isosurface is set to 0.006 e/Å³. For convenient comparison, we adopt a similar illustration of atoms as the reference. The primitive cell is illustrated with gray rectangles.

Reviewer #3 (Remarks to the Author):

The manuscript by Liu et al. presents spectroscopic investigation of the novel metallic stripe phase in NbSe₂ monolayer grown with molecular beam epitaxy under reduced Se flux. The authors focus on its most salient feature – narrow dip in the vicinity of the Fermi level – its depth and width exhibit intriguing quasi-1D spatial dependence. With the help of DFT calculations, this feature is ascribed to the Kondo state. The authors, thus, suggest that Se-deficient stripes in NbSe₂ could be the first experimental realization of the archetypal 1D Kondo model, where a variety of exotic excitations has been predicted.

The manuscript attempts to address challenging yet highly rewarding (if successful) problem of emergent magnetic orders in non-magnetic TMDC monolayers and its heterostructures. It clearly demonstrates novel and intriguing results, but I am not convinced that the evidences in favor of a Kondo state are compelling. I believe the manuscript is not suitable for publication in Nature Communications, but the material science aspect – stripe phase growth and characterization – is clearly suitable for a regular publication elsewhere.

Thank the referee for commenting our work as novel and intriguing. The referee's comments are insightful and constructive, which help to improve our manuscript. To prove the Kondo state of our system, we provide evidences from the line shape of the spectra, the evolution of Kondo peak width Γ with temperature, and the spatial oscillations of the Kondo resonance around the Kondo defect, which can exclude its alternative scenarios of SDW or disorder-induced pseudo gap. Our point-by-point responses to the referee's issues are below.

1. My major concern is the lack of clear identification of localized magnetic moments. The authors suggest that 'more isolated Nb atoms' could act as such moments based on DFT calculations. However, neither of predictions of these calculations are subject to further scrutiny or experimental testing. Moreover, I can clearly see that the bands in spin-resolved theoretical spectrum are rather dispersive, with small effective mass not immediately suggestive of localization.

Thanks for raising this issue. We agree with the referee that it is ideal to directly identify the localized magnetic moments. However, the system doesn't expect long-range magnetic order, and its spins are Kondo-screened at low temperature, making it challenging to experimentally verify the local moments in a monolayer crystal. Nevertheless, the Kondo resonance could unambiguously prove the local moments despite indirectly.

As for DFT calculations, we have carefully considered various possible structure models and found that the structure used in our manuscript gives the best agreement with experimental STM maps and also gives moment matrix. For the concern of localization, as also pointed out by the Reviewer #1. Please see our Response to the 5th comment of Reviewer #1. In short, first, the bandwidth of the orbital contributing to the magnetic moments in NbSe₂ stripe phase is comparable with that in SD phase (Fig. R2 (a) and (b)). Second, the bandwidth ratio of flat band and dispersive band of stripe-phase is also comparable with LiV₂O₄, a typical *d* electron Kondo lattice.

2. The assignment of the spectral dip to Kondo state is premature, in my opinion. For example, spin-

density wave order on a part of (spin-polarized!) Fermi surface pockets could account for a temperature-dependent ‘pseudo’-gap. Interestingly, the ratio of the gap to apparent T_c is within the expected range.

Thanks for raising this alternative scenario. We could exclude the possibility of SDW state in our system from the following aspects.

SDW represents a spatial modulation of ordered spin density. It doesn’t explicitly impose antiphase relation requirement in charge density (please see Ref. Nat. Commun. 4, 1596 (2013) doi: 10.1038/ncomms2592). However, our system indicates clear contrast reversal in charge density, conforming to the CDW, instead of the SDW, scenario. Furthermore, in actual experimental systems, the SDW gaps usually vanish abruptly at the SDW transition temperatures, as is exemplified in Ref. Nat. Commun. 5, 3711 (2014). This is again different to our case, where the spectral dip evolves continuously with temperature.

The Kondo resonance should be a Fano resonance, whose spectral shape, defined by the q -factor, is determined by the relative tunneling ratio between the two paths into Kondo states and conduction electrons. In our system, the localized moments and the conduction electrons are both inside the same monolayer. This makes the ratio between the two tunneling paths, namely the q factor, sensitive to local perturbations. Such variation of spectral shapes can be seen in Fig. 5 and Fig. R3, ranging from dip-like spectrum to asymmetric Fano line shapes. The spectra showing asymmetric Fano line shapes are distinct to a SDW gap or a disorder-induced gap, that are spectral dips. Moreover, we have measured a line spectra along a CDW chain that is in proximity to some region with inhomogeneity [Fig. R6(a)]. As seen in Fig. R6(b), while the line spectra at large bias range of $[-0.5, 0.8$ eV] are relatively uniform with well-formed CDW state, the low energy range of $[-80, 80$ meV] indicates prominent variations [Fig. R6(c)], reflecting the strong impact of local perturbation, presumably strain effect, to the q factor. In particular, selected spectrum exhibits a sharp resonance peak at the Fermi level, whose peak width is determined as 6.2 meV from the Fano fitting, similar to the anti-resonance width in Fig. 3a. And, temperature evolution of the resonance peak follows the behavior expected for a Kondo peak, unambiguously justifying its Kondo origin.

For clarity, Fig. R6 has been added to supplementary information as Fig. S8. A sentence starting with “In our system, ...” is added to Page 8.

Fig. R6. **a**, STM image ($V_b = -0.5$ V and $I_t = 10$ pA) of the stripe-phase NbSe₂ containing local inhomogeneity at the top region of the image. **b**, **c**, 2D conductance plot of different energy ranges taken along the white arrow in **a**. **d**, dI/dV spectra ($V_b = -80$ mV, $I_t = 150$ pA, $V_{\text{mod}} = 1$ mV) extracted from **c** marked with corresponding color lines. **e**, Temperature evolution of the dI/dV spectra measured at the same location of the stripe-phase NbSe₂ ($V_b = 80$ mV, $I_t = 200$ pA, $V_{\text{mod}} = 1$ mV). Red curves are Fano fittings to the spectra. **f**, Extracted Kondo peak width Γ (red dots) against temperature. The blue curve is a fitting to the data according to the Kondo model, yielding a Kondo temperature of $T_k \approx 54$ K.

3. Moreover, non-stoichiometric composition of the monolayer and potentially random positions of missing Se could suggest that the introduced disorder could facilitate the disorder-induced localization forming a pseudogap (see e.g. Gao et al., Nano Lett. 2020, 20, 9, 6299–6305), completely unrelated to Kondo physics. Neither of the above possibilities (2,3) were ruled out.

Thanks for this issue. The disorder-induced pseudo gap can be safely excluded from the following aspects. First, while the stripe-phase NbSe₂ in the present system originates from 1T-NbSe₂ with line defects, it is long-range ordered with a well-defined stoichiometry of Nb₃Se₅ without prominent disorder, as is evidenced from the ordered spectroscopic mappings in Figs. 2b-d. This is distinct to the disordered system of Se-substituted 1T- TaS₂ reported in Nano Lett. 2020, 20, 9, 6299–6305. Second, as is seen in our response to the above question, the Kondo state origin of the spectra is evidenced from its Fano line shape, the evolution of its peak width Γ with temperature, and the Kondo hole behavior of spatial oscillation behavior in its peak width near defects. Particularly, the Kondo resonance could exhibit a peak shape at certain region of the film (see Fig. R6), presumably due to the presence of local strain that tunes the tunneling matrix between the itinerant bands and the localized bands of spin moments. The peak is of Kondo origin, as is seen from its evolution of peak width with temperature. As such, we could exclude the possibility of pseudo gap scenario.

For clarity, we have added a sentence starting with “The disorder-induced pseudo gap...” to Page 7.

4. Use of contrast inversion as a hallmark of CDW remains controversial (see e.g. Spera et al., PRL 125, 267603 (2020) and discussions therein). Further evidence is highly desirable to support the CDW rather than a structural origin of the reconstruction within the chain.

Thanks for bringing this article to our attention. Actually, this article substantiates the use of contrast inversion as a hallmark of CDW state, as can be justified from the following aspects.

- (1) As stated in this article, contrast inversion in STM images near two edges of CDW gap hallmarks the CDW state. In 2D CDW systems such as NbSe₂, contrast reversal is in general not expected due to band structure effect or strong lattice distortions, as is described in the 4th paragraph of the mentioned article. Our system is a quasi-1D CDW system, which is devoid of the complexity of the 2D CDW system.
- (2) CDW gaps conventionally surpass the Fermi level. However, as is seen from Fig. 4(f) of that paper, the CDW gap of 1T-TiSe₂ is below the Fermi level. This results in “the absence of contrast reversal between opposite bias polarity STM images as would be expected in the classic Peierls description” (see the quoted sentence in Page 5 of that paper). Nevertheless, contrast reversal in STM images of two CDW gap edges is still valid to justify the CDW state, as is shown in the schematics of Figs. 4(d-f).

For clarity, we have cited the mentioned paper as reference 34 to substantiate the CDW state.

REVIEWER COMMENTS

Reviewer #1 (Remarks to the Author):

I thank the authors for carefully replying my comments. But I am still confused about the Kondo screening in this system and the theoretical calculations. I cannot recommend the publication of this manuscript. I have a few comments for the authors to consider.

(1) Without the magnetic field response, it is very difficult to assign the dip-like feature to be Kondo resonance. As also mentioned by Reviewer 3, there are many possibilities for dip-like feature near the Fermi level. For example, there is also a sharp dip-like feature in the 1H-TaS₂ layer (Fig. 3 in Nature 599, 582 (2021)), where they think that is the heavy-fermion hybridization gap. The authors also observe the peak-like feature at the Fermi level (as shown in Fig. S8d), does the peak-like feature split under strong external magnetic fields?

(2) In order to have the charge transfer effect between stripe-phase NbSe₂ and graphene, the authors apply ~3% strain to have ~100 mV energy shift. Does the strained NbSe₂ energetically stable? What is the role of strain for the charge transfer?

(3) The authors think that for the stripe-phase NbSe₂, the localized spin lattice and the itinerant electrons are in the same monolayer. The localized spin is contributed by the dx₂-y₂ orbital with ~0.35 eV bandwidth, and the conduction electron is from the dz₂ orbital with ~0.7 eV bandwidth. If this is the case, what is the critical bandwidth for forming the localized moment in this system? Is there clear feature for the “flat band” in the dI/dV spectrum in Fig. 2a? The authors claim that the ratio of dispersive band and flat band in stripe phase is comparable with that in LiV₂O₄. However, the bandwidth values for LiV₂O₄ (Refs. 40 and 41) are from theoretical calculations, and the authors should find the experimental values for the bandwidths in LiV₂O₄ to support their claim.

(4) As shown in Fig. 13a, the dx₂-y₂ band for forming the local moments is far from the Fermi level, how can the conduction electrons near the Fermi level screen the local moments to form the Kondo screening?

(5) I appreciate the authors' efforts for measuring more Kondo resonance oscillations adjacent to different atomic defects. Since the authors have got consistent data on several atomic defects, I would suggest the authors replace Fig. 5 with Fig. 15. This would be easier for the readers to see more STS data near the defects, and it could also make the conclusion about the Kondo resonance oscillation more convincing.

Reviewer #2 (Remarks to the Author):

My concerns on this manuscript have been properly addressed.

Reviewer #3 (Remarks to the Author):

The revised manuscript addresses some of the issues raised in my review. Unfortunately, the arguments put forward do not add up to a solid demonstration of Kondo lattice. I reiterate my previous assessment that the manuscript has very strong material science side, yet it is not suitable for publication in Nature Communications in its present form.

In particular, the authors suggest three evidences to support Kondo scenario: (i) Fano lineshape, (ii) evolution of the peak width with temperature and (iii) spatial oscillations of the peak width near defects. Out of these three, I find the last one most promising, presenting distinct physics and possible way to distinguish Kondo lattice from some kind of magnetic ordering like SDW. The most puzzling aspect is the apparent 4-period commensurability of the width modulation. How fast does the modulation decay in space? Is it symmetric with respect to the defect (before and after on the chain?). I am happy to reconsider, seeing more evidence towards Kondo origin of this feature, along with further support for origin of local moments.

Below are my follow-up comments to the authors replies.

Comment to Reply #1. The authors demonstrate d-band projected DFT spectra, suggesting possible quantitative routes to localization. I am surprised by the choice of band below -0.6 eV as a source of the localized moments, and would rather expect the one closer to the Fermi level. I suggest the authors provide spatial projections of the respective orbitals at different energy ranges, to ensure that they are really localized in space.

Comment to Reply #2. The authors dismiss the spin-density wave scenario based on the following arguments:

a) Lack of contrast inversion. Here, the authors misinterpret my question. I suggest that the SDW transition is responsible for opening of the ~ 20 -30 meV dip in the tunnelling spectra, as shown in Fig. 3a. The authors invoke the contrast inversion argument (shown in Fig. 2) at large energies, -0.35V and 0.15V, to dismiss the SDW scenario in favor of CDW, thus addressing the issue different from the one being raised.

b) Abrupt vanishing of SDW with temperature. The case cited by the authors has rather specific origin with magnetoelastic coupling between two order parameters (cf. Avci et al., Phys. Rev. B 83, 172503 (2011)) and associated first-order-like behavior. In contrast, there are archetypal examples in organic quasi-1D conductors (Bechgaard salts), where the well-established SDW phase transition is of the second order (cf. e.g. H. Yang, J.C. Lasjaunias, P. Monceau, J. Phys. Condens. Matter 11, 5083 (1999)).

c) Asymmetric Fano lineshape. Fano lineshape fitting is a tool – not a proof by itself. I would like to emphasize that the dip discussed is located on top of large-slope spectral curve or, even, a local maximum (cf. Fig. 2a, “broad band at 0.05 and 0.15 eV”). Local variations in this slope (Fig. R6c shows variations at 50 meV) could cause changes to the dip either.

Let us assume that it is the Fano q -factor that varies so strongly due to a redistribution between tunnelling channels caused by e.g. strain. Could the authors use DFT to demonstrate (not speculate) that such redistribution can actually occur – tunnelling to a specific orbital becomes more pronounced?

d) Robust tunnelling spectra in the CDW energy range, in contrast to the low-energy spatial inhomogeneity. This mismatch has a long history, especially in cuprates. If we limit ourselves to CDW physics, then inhomogeneities are known to induce in-gap states, as discussed in series of papers by Tütto and Zawadowsky, in particular, Phys. Rev. B 32, 2449 (1985). How does this help rule out the SDW scenario?

To summarize, I cannot see the unambiguous, smoking gun evidence for Kondo physics.

Reply to Review

Reviewer #1 (Remarks to the Author):

I thank the authors for carefully replying my comments. But I am still confused about the Kondo screening in this system and the theoretical calculations. I cannot recommend the publication of this manuscript. I have a few comments for the authors to consider.

Thank the reviewer for recognizing our efforts in replying the review comments. Below are our point-by-point responses to the remaining concerns.

(1) Without the magnetic field response, it is very difficult to assign the dip-like feature to be Kondo resonance. As also mentioned by Reviewer 3, there are many possibilities for dip-like feature near the Fermi level. For example, there is also a sharp dip-like feature in the 1H-TaS₂ layer (Fig. 3 in Nature 599, 582 (2021)), where they think that is the heavy-fermion hybridization gap. The authors also observe the peak-like feature at the Fermi level (as shown in Fig. S8(d)), does the peak-like feature split under strong external magnetic fields?

In Kondo lattice system, the exchange coupling among individual Kondo states opens a hybridization gap between the localized spin band and the itinerant band. In STM experiments, the interference between the two tunneling paths to the localized band and the itinerant band determines the shape of the Kondo lattice spectra. In the mentioned system of 1T/1H-TaS₂ reported in Nature 599, 582 (2021), itinerant band has a larger tunneling contribution to the tip in the 1H-TaS₂ layer, and is thus probed as a hybridization gap. On the other hand, in the case of 1T-TaS₂ layer, the heavy band contributes dominantly to the tunneling current, and is then probed as a peak feature. Strictly speaking, the Kondo lattice spectra should be described with the cotunneling model. Please see Fig. R5 for our numerical simulations on the variation of the Kondo lineshape with the tunneling ratio between the two channels. It's noted that the cotunneling model contains many parameters, making such fitting less straightforward. Thus, in actual Kondo lattice systems, the spectra for single ion Kondo state is frequently used to the fitting for simplicity (for instance, in Refs. PNAS 107, 10383 (2010), Nature 474, 362 (2011)).

In our system, the itinerant band and the heavy band are within the same layer, and their relative contributions to the tunneling current are tuned by local perturbations such as strain, making the Kondo lattice spectra vary. The dip-like feature in Fig. 3a and the peak-like feature in our Fig. S8(d) should both split under strong external magnetic field. However, as is mentioned in our last round of reply, the Zeeman splitting energy for a Kondo peak is given as $2g\mu_B B$, which is 2.32 meV under a 10 T magnetic field for $g = 2$. Such Zeeman splitting energy is much smaller than our spectral width of the dip and the peak, i.e. ~ 8 meV. Namely, to observe the expected Kondo spectra splitting, a magnetic field as large as ~ 40 T is needed, which is beyond the maximum field strength of our instrument. We would like to mention that in many Kondo lattice systems, the applied magnetic field cannot split the Kondo spectra (see examples in Fig. R1), because of the larger Kondo peak width than the Zeeman splitting energy as mentioned above.

For clarity, we have added a sentence "Note that our applied magnetic field cannot split the Kondo peak, because of the larger Kondo peak width than the Zeeman splitting energy." to the caption of Fig. S8.

Fig. R1. Kondo resonance peak under magnetic field. Magnetic field dependences of the Kondo resonance peak in Pb intercalated 1T-TaS₂ (a) [quoted from Ref. Nat. Commun. 13, 2156 (2022)], and kagome magnet Mn₃Sn (b) [quoted from Ref. PRL 125, 046401 (2020)].

(2) In order to have the charge transfer effect between stripe-phase NbSe₂ and graphene, the authors apply ~3% strain to have ~100 mV energy shift. Does the strained NbSe₂ energetically stable? What is the role of strain for the charge transfer?

The strained stripe-phase NbSe₂ monolayer is energetically stable as no obvious structural distortion is further observed in the strained lattice. As can be seen in the Fig. R2, strain can promote the charge transfer effect between the graphene and NbSe₂ stripe-phase. The shift caused by the pure charge transfer effect is ~ 70 mV and ~3% strain increases the energy shift further by ~ 30 mV. This could result from smaller work function (energy difference between vacuum and Fermi level) of the strained stripe-phase, thus larger charge transferring from NbSe₂ stripe-phase to the substrate.

For clarity, we have added a phrase “or strain induced by sample inhomogeneity” to Page 10, and modified Fig. S11.

Fig. R2. DOS of monolayer stripe-phase NbSe₂ under different condition.

(3) The authors think that for the stripe-phase NbSe₂, the localized spin lattice and the itinerant electrons are in the same monolayer. The localized spin is contributed by the dx²-y² orbital with ~0.35 eV bandwidth, and the conduction electron is from the dz² orbital with ~0.7 eV bandwidth. If this is the case, what is the critical bandwidth for forming the localized moment in this system? Is there clear feature for the “flat band” in the dI/dV spectrum in Fig. 2a? The authors claim that the ratio of dispersive band and flat band in stripe phase is comparable with that in LiV₂O₄. However, the bandwidth values for LiV₂O₄ (Refs. 40 and 41) are from theoretical calculations, and the authors should find the experimental values for the bandwidths in LiV₂O₄ to support their claim.

As for the critical bandwidth, we note that while from a DFT calculation aspect it would be difficult to determine the critical bandwidth for localized magnetic moments to reach Kondo lattice, a two-band model has been proposed to explain the heavy Fermion behavior of LiV₂O₄ (Phys. Rev. B 62, 4403 (2000)). This research suggests that the bandwidth may not be the critical influence for the emerging Kondo resonance of *d* electron systems. Instead, the inter-band Coulomb repulsion plays a vital role, which may also matter in our system. The experimental signature of the flat band cannot be observed as it is in the deep valence band and mixing with other valence bands.

As for the experimental values for the bandwidths, we’ve only found photoemission spectra experiments, which only address the energy range of ~50 mV around the Fermi level (Phys. Rev. Lett 96, 026403 (2006)). Though theoretical calculation (Phys. Rev. Lett 83, 364 (1999)) agrees well within the small energy range, it may not be enough to make a detailed comparison regarding different bandwidth of *d* orbitals. Besides, it would also be a challenge to resolve the band contribution from different orbitals experimentally.

(4) As shown in Fig. 13a, the dx²-y² band for forming the local moments is far from the Fermi level, how can the conduction electrons near the Fermi level screen the local moments to form the Kondo screening?

It’s indeed that dx²-y² band is far from the Fermi level, but can be screened by the conduction electrons. In Kondo lattice systems, the orbital bands forming the local moments are conventionally far from the Fermi level, but can be screened by the itinerant conduction electrons via spin-flip scattering, which involves a virtual hopping process. Here, we show a typical example of Kondo lattice in Mn₃Sn (Ref. PRL 125, 046401 (2020)), where the flat band is far from the Fermi level, as is shown in Fig. R3(c) (the flat bands are marked with black arrows) and the schematic drawing in Fig. R3(b).

For clarity, we have added a sentence “Although these *d* bands that contribute to the major localized magnetic moments are far below the conducting bands holding itinerant electrons (Fig. S14) in terms of their energy level positions, these moments can still be effectively screened by the itinerant conduction electrons via spin-flip scattering, which involves a virtual hopping process [44].” to Page 11. And cited the mentioned paper as reference 44.

Fig. R3. **a**, Schematic depicting a Kondo lattice formed by the coupling between periodic localized states (red arrows) and itinerant conduction electrons (blue arrows). **b**, DOS spectrum of the Kondo lattice, where a Kondo resonance (dark blue) at E_F is generated by the many-body coupling of the localized flatband (red) and itinerant conduction band (shaded blue). **c**, Normalized bulk band structure of Mn_3Sn in the conformal Brillouin zone showing low energy flatbands (black arrows) [quoted from Ref. PRL 125, 046401 (2020)].

(5) I appreciate the authors' efforts for measuring more Kondo resonance oscillations adjacent to different atomic defects. Since the authors have got consistent data on several atomic defects, I would suggest the authors replace Fig. 5 with Fig. S15. This would be easier for the readers to see more STS data near the defects, and it could also make the conclusion about the Kondo resonance oscillation more convincing.

Thanks for this nice suggestion. Following the reviewer's suggestion, we have replaced the original Fig. 5 with Fig. S15.

Reviewer #2 (Remarks to the Author):

My concerns on this manuscript have been properly addressed.

Thanks for recommending to publish our work in Nature Communications.

Reviewer #3 (Remarks to the Author):

The revised manuscript addresses some of the issues raised in my review. Unfortunately, the arguments put forward do not add up to a solid demonstration of Kondo lattice. I reiterate my previous assessment that the manuscript has very strong material science side, yet it is not suitable for publication in Nature Communications in its present form.

In particular, the authors suggest three evidences to support Kondo scenario: (i) Fano lineshape, (ii) evolution of the peak width with temperature and (iii) spatial oscillations of the peak width near defects. Out of these three, I find the last one most promising, presenting distinct physics and possible way to distinguish Kondo lattice from some kind of magnetic ordering like SDW. The most

puzzling aspect is the apparent 4-period commensurability of the width modulation. How fast does the modulation decay in space? Is it symmetric with respect to the defect (before and after on the chain?). I am happy to reconsider, seeing more evidence towards Kondo origin of this feature, along with further support for origin of local moments.

Thank the reviewer for the recognition that some of the issues have been addressed, and the comment that one of the evidences is promising to justify the distinct physics of the Kondo lattice. Below are our point-by-point responses to the reviewer's concerns.

- (1) The apparent 4-period commensurability of the width modulation is related to the Fermi wavelength of the itinerant band, as depicted in Page 12 of the main text. The spatial oscillation of the resonance width is a signature for the Kondo lattice, because of the absence of local moment at defect as a Kondo hole. As mentioned in Ref. PNAS 108, 18233 (2011), the Kondo hole induces spatial oscillation of the local hybridization strength between the spin lattice band and the itinerant band. Such hybridization strength is associated with the Kondo resonance width, as determined experimentally. The wave-vector characterizing such oscillation period is given as $2k_F^c$, where k_F^c is the Fermi wave vector of the itinerant band. Our DFT calculation gives $2k_F^c \approx 0.75$ reciprocal lattice units (r.l.u.), which corresponds to a wave vector Q^* of about 0.25 r.l.u. in the extended zone scheme. This conforms to a period of about 4 times the CDW period in space.
- (2) To evaluate the modulation decay of the Kondo resonance, we show in Figs. R4(a-f) the dI/dV spectra taken along the blue arrows in the inset images and their Kondo resonance width Γ extracted from the Fano fitting. Obviously, all the three data sets in Fig. R4(a-f) exhibit spatial oscillations in Γ along the direction of the stripe and decay gradually away from the defects. We fitted the oscillation decay with an exponentially decayed sinusoidal function, i.e. $\Gamma(r) = Ae^{-\sigma r} \sin(\omega r + \varphi) + b$. The fitting gives a similar decay length of $\sim 8.3 \pm 0.6$ nm and an oscillation period $\sim 3.9 \pm 0.6$ nm for all the three data sets. Yet, while the modulation decay is evident, the complete disappearance of the oscillation was not observed due to the limited films size of our sample.
- (3) To examine whether such oscillation is symmetric relative to the defect, we take dI/dV spectra along the stripe traversing a defect (Figs. R4g,h). The Kondo resonance width Γ that is extracted from the Fano fitting indicates spatial oscillations, whose oscillation periods are similar with symmetric distribution on both sides of the defect.

Fig. R4. a, c, e, g, Spatial oscillation of the Kondo peak width Γ (red dots) near defects along the strip direction, that are extracted by fitting the dI/dV spectra in **b, d, f, h** respectively. The blue curves are fittings to the data with an exponentially decayed sinusoidal function. Inset STM images show the stripe phase NbSe_2 with marked defects (black arrow) and the spectral locations (blue arrow). **b, d, f, h,** The dI/dV spectra taken along the blue arrows in the inset images of **a, c, e, g**. The red curve in **d** shows a typical Fano fit to the spectrum.

We would like to emphasize that, besides the Kondo hybridization strength oscillation, the evolution of the Kondo lineshape under local perturbations is also a very strong evidence for the coherent Kondo lattice behavior. In our system, the localized moments and the conduction electrons are both inside the same monolayer. This makes the ratio between the two tunneling paths, namely the q factor that characterizes the Kondo lineshape, sensitive to local perturbations. We provided such data in Fig. S8 of our last reply. As shown in Fig. S8 in the Supplemental Information, the CDW state is uniform adjacent to a region with inhomogeneity (Fig. S8b), demonstrating the overall electronic structure of our system is unchanged. However, the Kondo lineshape varies strongly (Fig. S8c). Particularly, a narrow spectral peak (Fig. S8d, black curve) and a spectral gap with two enhanced peaks at the gap edges (Fig. S8d, red curve) are seen. The temperature evolution of the narrow peak follows nicely to the behavior of the Kondo peak, rigorously demonstrating its Kondo origin (Figs. S8e,f).

To reproduce the observed variation of the spectral lineshape, we modelled the tunneling spectra with the cotunneling model that has been well-established for the Kondo lattice system (Ref. Phys. Rev. Lett 103, 206402 (2009)). This model is more complex than the simple Fano lineshape fitting, but is more precise to describe the Kondo lattice spectra. The modelling is based on a electron-like conduction band $E_k^c = -2t(\cos k_x + \cos k_y) + \mu$ (green band in Fig. R5a) and a heavy flat band $E_k^f = -2\chi_0(\cos k_x + \cos k_y) + \varepsilon_0^f$ (red band in Fig. R5a) near the Fermi energy (t : nearest neighbor hopping of the conduction electrons; μ : chemical potential; χ_0 : nearest site spin correlation; ε_0^f : the position of the heavy band respect to Fermi energy). The coherent Kondo

screening gives two heavy fermion bands: $E_k^\pm = \frac{E_k^c + E_k^f}{2} + \sqrt{\left(\frac{E_k^c - E_k^f}{2}\right)^2 + v^2}$; where v is the

hybridization amplitude between the conduction and heavy flat bands. The differential conductance dI/dV is given by $dI/dV \propto \sum_{i,j=1}^2 [tI m G(k, \omega)]_{ij}$; where t depicts the tunneling ratio between the conduction band and the flat band, and $G(k, \omega)$ is the full Green's function describing the hybridization between the above two bands, whose details are seen in Ref. Nature 486, 201 (2012). Based on this model with the same band structure shown in Fig. R5a, we simulate three dI/dV spectra with three selected values of t_f/t_c of 0.15, 0.03 and 0.002, where t_f and t_c are the tunneling amplitudes to the heavy and light bands, respectively. The simulated spectra shown in Fig. R5b well reproduce the experimental spectra in Fig. S8d and Fig. 3, demonstrating those spectra are all from Kondo resonances, but with different tunneling ratio between the heavy and electron bands, presumably due to the influence of strain.

For clarity, we have added the above discussion to Supplementary Note 1. Two sentences starting with “and a Kondo gap ...” are added to Page 8. Four sentences starting with “The oscillations decay from defects...” are added to Page 12. Figs. R4g,h and R5 are added to Supplementary Information as Figs. S16 and S17.

Fig. R5. a, Dispersion of the conduction light (dashed green) and flat (dashed red) electronic bands and the hybridized heavy fermion bands (solid black) computed for $t = 200\text{meV}$, $\mu = 2t$, $\chi_0 = 0.05t$, $\varepsilon_0^f = 0.1t$, $v = 0.5t$, $\gamma_f = 0.02t$. Here γ_f is the scattering rate of the heavy flat band. **d**, Differential conductance computed for the heavy fermion bands of a, using different values of t_f/t_c . The value of v is $0.5t$ for the top and middle curve, and $0.52t$ for the bottom curve.

Below are my follow-up comments to the authors replies.

Comment to Reply #1. The authors demonstrate d-band projected DFT spectra, suggesting possible quantitative routes to localization. I am surprised by the choice of band below -0.6 eV as a source

of the localized moments, and would rather expect the one closer to the Fermi level. I suggest the authors provide spatial projections of the respective orbitals at different energy ranges, to ensure that they are really localized in space.

We thank the comment from the referee. As shown in the Fig. R6, we've provided spatial projections of magnetic moments at different energy ranges. In the energy range 1(-0.65~-0.4 eV) we can see clearly spin \uparrow moments with $d_{x^2-y^2}$ character situated in the shallow blue Nb sites. In the energy range 2(-0.4~-0.15 eV) the magnetic moments with opposite spin are observed at Nb atoms of both edges of stripe except for the shallow blue sites. In the energy range 3(-0.1~-0.12 eV), which crosses the Fermi level, spin \uparrow moments are mainly distributed in Nb atoms of right edge of stripe, also including shallow blue Nb atoms. Comparing **b** with **d**, magnetic moments ~ 0.5 eV below the Fermi level could be well described as local moments while magnetic moments near the Fermi level show distinctive one-dimensional character but could not be described as localized. Besides, we notice that in the Kondo lattice systems, the orbital bands forming the local moments are conventionally far from the Fermi level, but can be screened by the itinerant conduction electrons via spin-flip scattering, which involves a virtual hopping process. A typical example for such a Kondo lattice system is Mn_3Sn (please see our reply to the 4th comment of Referee 1).

For clarity, a sentence "Although these d bands that contribute to the major localized magnetic moments are far below the conducting bands holding itinerant electrons (Fig. S14) in terms of their energy level positions, ..." is added to Page 11, and Fig. R6 is added to Supplementary Information as Fig. S14.

Fig. R6 Spatial distribution of magnetic moments of NbSe₂ stripe-phase at different energy range. **a**, DOS difference (spin \uparrow - spin \downarrow). **b**, **c**, and **d** magnetic moments at corresponding energy range 1(-0.65~-0.4 eV), 2(-0.4~-0.15 eV) and 3(-0.1~-0.12 eV). The iso-surface is set to $1 \times 10^{-3} \text{ e}/\text{\AA}^3$ and orange (blue) isosurface represents spin \uparrow (spin \downarrow) magnetic moments. The Nb atoms proposed to provide local moments for the Kondo lattice are illustrated with shallow blue.

Comment to Reply #2. The authors dismiss the spin-density wave scenario based on the following arguments:

a) Lack of contrast inversion. Here, the authors misinterpret my question. I suggest that the SDW transition is responsible for opening of the ~ 20 -30 meV dip in the tunnelling spectra, as shown in

Fig. 3a. The authors invoke the contrast inversion argument (shown in Fig. 2) at large energies, -0.35V and 0.15V, to dismiss the SDW scenario in favor of CDW, thus addressing the issue different from the one being raised.

We apologize for misinterpreting the reviewer's question.

b) Abrupt vanishing of SDW with temperature. The case cited by the authors has rather specific origin with magnetoelastic coupling between two order parameters (cf. Avci et al., Phys. Rev. B 83, 172503 (2011)) and associated first-order-like behavior. In contrast, there are archetypal examples in organic quasi-1D conductors (Bechgaard salts), where the well-established SDW phase transition is of the second order (cf. e.g. H. Yang, J.C. Lasjaunias, P. Monceau, J. Phys. Condens. Matter 11, 5083 (1999)).

Thank the reviewer for providing us the helpful references. We realize the SDW is of second order and our former argument is inappropriate. Nevertheless, while the Kondo state cannot be reproduced from first-principles calculations, the SDW state should in principle be captured from first-principles calculations. Although without knowing the period and of SDW and arrangement of moments we could not exclude all SDW possibilities, we inspect two possible SDWs from first-principles calculations. As shown in Fig. R7, we see no obvious DOS reduction at the Fermi level for SDW1 but strong DOS reduction for SDW2. However, no dips or peaks \sim several mV can be observed in the shifted Fermi level, indicating the small dips and peaks around Fermi level found in experiments may not be attributed to SDWs.

For clarity, we have added the above discussion to Supplementary Note 3. Fig. R7 is added to Supplementary Information as Fig. S19.

Fig. R7. DFT-calculated two possible SDW states. **a**, DOS of SDW states comparing with the original CDW state. The Fermi level is set to zero and the dashed line indicate the shifted Fermi level due to substrate. **b** and **c**, two possible SDWs in the NbSe₂ stripe-phase. The local moments with spin \uparrow (\downarrow) are illustrated with orange (green) color.

c) Asymmetric Fano lineshape. Fano lineshape fitting is a tool – not a proof by itself. I would like to emphasize that the dip discussed is located on top of large-slope spectral curve or, even, a local maximum (cf. Fig. 2a, “broad band at 0.05 and 0.15 eV”). Local variations in this slope (Fig. R6(c) shows variations at 50 meV) could cause changes to the dip either.

We agree that there is a large slope spectra curve. As is pointed out by the reviewer, there is a local maximum at 50 meV and 150 meV in Fig. 2a, but they are still far from the Kondo spectra at the Fermi level, whose spectra range is within [-25, 25] meV. In other words, the variation of the slope spectra is in a larger energy range, and cannot change the Kondo lineshape.

In order to quantitatively evaluate the impact of the background slope, we omitted the energy range of [-40, 40] meV of Fig. S8d and fitted the remaining spectra with a polynomial function (dashed line in Fig. R8b). Such fitted background also reasonably reproduces the background spectra in Fig. R8a, except the small bump centered around 50 mV. Then, the fitted background is subtracted from the two experimental spectra to generate the intrinsic Kondo lineshapes (Fig. R8, black curves). Evidently, the *intrinsic* Kondo peak in Fig. R8a and the gapped Kondo peak in Fig. R8b are seen without prominent impact of the background spectra. Those Kondo lineshapes could be nicely reproduced with the cotunneling model (red curves in Fig. R8), well proving their characteristic features to Kondo lattice spectra.

For clarity, we combined Fig. R8 into Fig. S8d and modified corresponding figure captions.

Fig. R8. Kondo spectra with different lineshapes and their fitting with cotunneling model. The spectra in blue are from Fig. S8d. The dashed lines are the fitted background spectrum from **b**. The spectra in black are from the spectra in blue after subtracting the fitted background. The spectra in red are calculated from the cotunneling model with parameters $t_f/t_c = 0.025$, $\gamma_f = 0.027t$ for **a**, and $t_f/t_c = -0.01$, $\gamma_f = 0.021t$ for **b**.

Let us assume that it is the Fano q-factor that varies so strongly due to a redistribution between tunnelling channels caused by e.g. strain. Could the authors use DFT to demonstrate (not speculate) that such redistribution can actually occur – tunnelling to a specific orbital becomes more pronounced?

To unveil the strain effect on the redistribution of tunneling channels, the tunneling spectra is simulated by calculating local DOS (LDOS) in spheres $\sim 4 \text{ \AA}$ above Nb atoms, as shown in Fig. R9. We note that in our DFT simulation the one-particle mean-field approach could not take physics of paired Kondo singlet into account. However, the qualitative analysis could also provide insights into the large variation of tunneling channels of the stripe-phase NbSe₂ monolayer. The strength of local moments at $\sim 0.5 \text{ eV}$ below the Fermi level hasn't been influenced strongly (red lines) by the strain, However, strain strongly affects the LDOS of conduction electrons (blue lines) at the shifted Fermi level. For example, by applying 3% tensile strain, the LDOS is strongly suppressed at the shifted

Fermi level (only $\sim 1/3$ of the unstrained one). This indicates the tunneling from the tip to the conduction electrons could also be strongly suppressed. Overall, the conduction electrons can vary strongly when strain is applied.

For clarity, we have added the above discussion to Supplementary Note 2. Fig. R9 is added to Supplementary Information as Fig. S18. A sentence “Interestingly, the DFT calculations also reveal that the tunneling amplitude to the itinerant bands could be sensitive to strain (Supplementary Note 2).” is added to Page 8.

Fig. R9. DFT-calculated local DOS with/without 3% tensile strain. The spin- \uparrow LDOS is calculated in a sphere above Nb atoms with local magnetic moments (shallow blue sites in Fig. R6). The spin- \downarrow LDOS is calculated in a sphere above Nb atoms neighboring to shallow blue Nb atoms, which provide conduction electrons at the shifted Fermi level (dash line).

d) Robust tunnelling spectra in the CDW energy range, in contrast to the low-energy spatial inhomogeneity. This mismatch has a long history, especially in cuprates. If we limit ourselves to CDW physics, then inhomogeneities are known to induce in-gap states, as discussed in series of papers by Tütto and Zawadowsky, in particular, Phys. Rev. B 32, 2449 (1985). How does this help rule out the SDW scenario?

Thanks for mentioning this article. As is theoretically predicted in Phys. Rev. B 32, 2449 (1985), the coupling of inhomogeneities with CDW will produce a pair of in-gap states that are symmetrical with respect to the center of the CDW gap, and they are typically located at the CDW gap edges. Such prediction is not observed in our case. In our experiments, the CDW gap center is located at -0.1 V, while the low-energy features we observed in Fig. S8 are located near the Fermi surface, which are not symmetrical with respect to the CDW gap center. Therefore, the low-energy characteristics we observe are distinct to in-gap states induced by CDW inhomogeneities. Further, as observed in Fig. S8, the modified low-energy features that are affected by local perturbations can be reproduced with the cotunneling model by changing the ratio of the two tunneling paths (Fig. R5), which confirms that the low-energy features are specifically Kondo resonances.

For clarity, we have added the above discussion to Supplementary Note 3, and cited the mentioned paper as Supplementary Reference 3.

To summarize, I cannot see the unambiguous, smoking gun evidence for Kondo physics.

Here, we provided (i) the spatial oscillation of the Kondo line width, (ii) the Kondo lineshape changes reproduced with the cotunneling model, (iii) temperature dependent evolution of the Kondo lineshape, as evidences for the Kondo lattice behavior. Those observations conform to the establishment of coherent Kondo lattice, but are incompatible with other scenarios such as SDW. For clarity, we have added a sentences starting with “Overall, based on...” to Page 13.

We hope the reviewer would recognize the validity of those evidences and our efforts in the additional studies. The reviewer’s suggestion for publication of our work in Nat. Commun. would be highly appreciated.

REVIEWERS' COMMENTS

Reviewer #1 (Remarks to the Author):

First of all, I thank the authors for the detailed replies to my comments. But I am still not convinced about the interpretation for the dip-like feature as a Kondo resonance in the stripe-phase 1T-NbSe₂. In comparison with the sharp-peak feature at the Fermi level, without magnetic-field response, it is even more difficult to assign a dip-like feature as a Kondo resonance. On the other hand, the band for forming the localized moments in this system is not flat at all (with ~0.35 eV bandwidth), and it is located far below the Fermi level. If this kind of band can form localized moments and induce Kondo screening, there would be Kondo resonances in many materials. Although I still cannot recommend the publication of this manuscript in Nature Communications, I will leave the editor and the other reviewers to decide whether this manuscript should be published.

Reviewer #3 (Remarks to the Author):

I have reviewed the revised manuscript and thank the authors for carefully and diligently addressing the raised issues. I highly value the additional data and calculations, which make the main message consistent and robust. They further make the manuscript accessible to broader community outside the Kondo physics one. The material system itself is equally interesting and, together with the presented first-principle characterization, is likely to be interesting to the strongly-correlated community. I am happy to recommend the paper for publication in Nature Communications.

I have minor comments which I suggest the authors to consider to improve the appeal of the manuscript to the broad context of readers:

1. It would be valuable to add the *ab initio* susceptibility for the SDW formation as a function of momentum, such that the potential ordering vectors could be identified. It could happen that additional strain could favor spin ordering.
2. In the last paragraph before the summary the authors write: "Overall, based on (i) the spatial oscillation of the Kondo linewidth, (ii) the changes of Kondo lineshape reproduced with the cotunneling model, (iii) temperature-dependent evolution of the Kondo lineshape, the Kondo lattice behavior can be well identified, excluding other scenarios (Supplementary Note 3)." I would suggest to change the wording to: "...the Kondo lattice behavior can be well identified, making alternative charge- or spin-ordering scenarios unlikely". I believe this aligns more with what the authors wrote in the reply to my comments.

3. Whereas the Kondo lattice scenario is plausible, part of the picture that supports it (e. g. the origin of local spins) is not directly confirmed experimentally and is suggested by ab initio calculations. In the summary, the authors write: “Those observations are substantiated with DFT calculations, which reproduce the CDW order coexisting with an ordered lattice of local moments, as well as the 1D character of the electronic states.” I believe, “reproduce” is too strong of the statement here and suggest to change it to “suggest”, as there is no separate knowledge on e.g. the precise atomic positions in the unit cell.

REVIEWERS' COMMENTS

Reviewer #1 (Remarks to the Author):

First of all, I thank the authors for the detailed replies to my comments. But I am still not convinced about the interpretation for the dip-like feature as a Kondo resonance in the stripe-phase 1T-NbSe₂. In comparison with the sharp-peak feature at the Fermi level, without magnetic-field response, it is even more difficult to assign a dip-like feature as a Kondo resonance. On the other hand, the band for forming the localized moments in this system is not flat at all (with ~0.35 eV bandwidth), and it is located far below the Fermi level. If this kind of band can form localized moments and induce Kondo screening, there would be Kondo resonances in many materials. Although I still cannot recommend the publication of this manuscript in Nature Communications, I will leave the editor and the other reviewers to decide whether this manuscript should be published.

Thank the reviewer for the additional comments. As mentioned in our last reply, **similar to many other Kondo systems**, the Kondo resonance width in our case is too large to induce an observable Zeeman splitting under the available magnetic field strength. Nevertheless, we have provided rigorous experimental evidences for the Kondo resonance, including (i) the spatial oscillation of the Kondo linewidth, (ii) the changes of Kondo lineshape reproduced with the cotunneling model, (iii) temperature-dependent evolution of the Kondo lineshape.

Whether a band is localized enough to induce localized moments depends on not only the absolute value of the band width but also the Coulomb repulsion energy. Localized moments form upon the Coulomb repulsion energy is larger than the band width. For Kondo systems, the Kondo screening temperature is determined by not only the energy of the localized orbital, but also the coupling strength between the local moments and itinerant electrons (see Ref. J. Phys.: Condens. Matter 21, 053001(2009)). Thus, in many Kondo systems, the localized orbitals are far away from the Fermi level. For instance, the localized f-orbital in the Kondo lattice system of Ce films is located 1.9 eV below the Fermi level (Ref. Nat Commun 12, 2520 (2021)).

For clarity, we have added a sentence “The Kondo resonance width in our case is too large to induce an observable Zeeman splitting under the available magnetic field strength.” to Page 9 of the main text.

Reviewer #3 (Remarks to the Author):

I have reviewed the revised manuscript and thank the authors for carefully and diligently addressing the raised issues. I highly value the additional data and calculations, which make the main message consistent and robust. They further make the manuscript accessible to broader community outside the Kondo physics one. The material system itself is equally interesting and, together with the presented first-principle characterization, is likely to be interesting to the strongly-correlated community. I am happy to recommend the paper for publication in Nature Communications.

Thank the reviewer for recommending to publish our work in Nature Communications. Below are

our point-by-point responses to the additional minor comments.

I have minor comments which I suggest the authors to consider to improve the appeal of the manuscript to the broad context of readers:

1. It would be valuable to add the ab initio susceptibility for the SDW formation as a function of momentum, such that the potential ordering vectors could be identified. It could happen that additional strain could favor spin ordering.

Thanks for your helpful suggestion. We've added the calculation results and related discussions of the susceptibility to our Supplementary Note 3 and Supplementary Fig. 20.

2. In the last paragraph before the summary the authors write: "Overall, based on (i) the spatial oscillation of the Kondo linewidth, (ii) the changes of Kondo lineshape reproduced with the cotunneling model, (iii) temperature-dependent evolution of the Kondo lineshape, the Kondo lattice behavior can be well identified, excluding other scenarios (Supplementary Note 3)." I would suggest to change the wording to: "...the Kondo lattice behavior can be well identified, making alternative charge- or spin-ordering scenarios unlikely". I believe this aligns more with what the authors wrote in the reply to my comments.

Following the reviewer's suggestion, we have changed the wording accordingly in Page 9 of the main text.

3. Whereas the Kondo lattice scenario is plausible, part of the picture that supports it (e. g. the origin of local spins) is not directly confirmed experimentally and is suggested by ab initio calculations. In the summary, the authors write: "Those observations are substantiated with DFT calculations, which reproduce the CDW order coexisting with an ordered lattice of local moments, as well as the 1D character of the electronic states." I believe, "reproduce" is too strong of the statement here and suggest to change it to "suggest", as there is no separate knowledge on e.g. the precise atomic positions in the unit cell.

Following the reviewer's suggestion, we have changed the wording of "reproduce" to "suggest" in the summary.